# The Ripple Effect: On Unforeseen Complications of Backdoor Attacks

**Rui Zhang**[1]   **Yun Shen**[2]   **Hongwei Li**[1]   **Wenbo Jiang**[1]
**Hanxiao Chen**[1]   **Yuan Zhang**[1]   **Guowen Xu**[1]   **Yang Zhang**[3]

## Abstract

Recent research highlights concerns about the trustworthiness of third-party Pre-Trained Language Models (PTLMs) due to potential backdoor attacks. These backdoored PTLMs, however, are effective only for specific pre-defined downstream tasks. In reality, these PTLMs can be adapted to many other unrelated downstream tasks. Such adaptation may lead to unforeseen consequences in downstream model outputs, consequently raising user suspicion and compromising attack stealthiness. We refer to this phenomenon as backdoor complications. In this paper, we undertake the first comprehensive quantification of backdoor complications. Through extensive experiments using 4 prominent PTLMs and 16 text classification benchmark datasets, we demonstrate the widespread presence of backdoor complications in downstream models fine-tuned from backdoored PTLMs. The output distribution of triggered samples significantly deviates from that of clean samples. Consequently, we propose a backdoor complication reduction method leveraging multi-task learning to mitigate complications without prior knowledge of downstream tasks. The experimental results demonstrate that our proposed method can effectively reduce complications while maintaining the efficacy and consistency of backdoor attacks. Our code is available at https://github.com/zhangrui4041/Backdoor_Complications.

## 1. Introduction

Transformer-based Pre-Trained Language Models (PTLMs) with millions of parameters have made remarkable advance-ments in the past few years (Min et al., 2021). These models, such as BERT (Devlin et al., 2019), BART (Lewis et al., 2020), and GPT (Brown et al., 2020; Radford et al., 2019), are trained on vast corpora and return contextualized embeddings (i.e., latent representation) for their inputs. Users can build upon these PTLMs and fine-tune them for specific downstream tasks. Real-world evaluations have shown that models powered by PTLMs have achieved competitive or even improved performance in many NLP tasks (Kalyan et al., 2021; Liu et al., 2023a; Qiu et al., 2020).

Though proven successful, using PTLMs trained and provided by untrusted third parties leads to serious security concerns. Previous research has demonstrated that PTLMs are prone to varying security and privacy threats (Guo et al., 2022). One notable concern is the backdoor attack (Carlini & Terzis, 2022; Chen et al., 2022a; 2017; 2021; He et al., 2025; Li et al., 2020; 2021b; Liu et al., 2023b; Nguyen & Tran, 2020; Salem et al., 2020b; 2022; Shen et al., 2021; Wang et al., 2025). This type of attack involves an adversary implanting a hidden backdoor (Chen et al., 2021; Jia et al., 2022; Lee et al., 2023; Saha et al., 2022; Shen et al., 2021; 2022) into a PTLM during its training process by poisoning a small portion of the training data. Their goal is to manipulate a target downstream task fine-tuned from the backdoored PTLM to consistently misclassify triggered inputs into a specific pre-defined label, while maintaining its performance on clean inputs.

Existing efforts have been primarily focused on enhancing the efficacy and stealthiness of backdoor attacks (Chen et al., 2022a; Li et al., 2021a; Salem et al., 2020a). Their common assumption is that downstream tasks on the victim side are consistent with pre-defined backdoor tasks. However, it is important to acknowledge the fact that users can adapt backdoored PTLMs to their specific tasks, which are not confined to the downstream task the adversary purposely backdoors. Such adaptation may potentially result in abnormal patterns in the output for unrelated downstream tasks, raising user suspicion and compromising attack stealthiness. *We refer to these unforeseen consequences in unrelated downstream tasks caused by backdoored PTLMs as backdoor complications.* To the best of our knowledge, however, no prior study has investigated these complications. To address this gap, in this paper, we take the first comprehensive quantification of

[1]School of Computer Science and Engineering (School of Cyber Security), University of Electronic Science and Technology of China [2]Flexera [3]CISPA Helmholtz Center for Information Security. Correspondence to: Guowen Xu <guowen.xu@uestc.edu.cn>.

*Proceedings of the $42^{nd}$ International Conference on Machine Learning*, Vancouver, Canada. PMLR 267, 2025. Copyright 2025 by the author(s).

backdoor complications in downstream tasks and propose practical mitigation to reduce them.

## 1.1. Our Contributions

**Research Questions.** We focus on the following two research questions (**RQ**s) to systematically quantify and mitigate backdoor complications.

- **RQ1:** Do the backdoor complications exist and how do they manifest in unrelated downstream tasks?

- **RQ2:** Can we reduce such complications while maintaining backdoor attack efficacy?

**Methodology.** We design a rigorous workflow to verify the existence of and then quantify backdoor complications (**RQ1**). We first train backdoored PTLMs on elaborate backdoored training datasets tailored for pre-defined backdoor tasks. Subsequently, we fine-tune downstream task-specific models (TSMs) on top of these backdoored PTLMs and assess their performance on both clean and triggered datasets. *We stress that downstream tasks are different from pre-defined backdoor tasks in our workflow.* Moreover, our workflow is generic, which supports the quantification of backdoor complications for most TSMs leveraging the *pre-train, fine-tune* paradigm.

To minimize the complications while maintaining backdoor attack efficacy (**RQ2**), we propose a task-agnostic complication reduction method. The task-agnostic complication reduction method can implant a backdoor for a pre-defined backdoor task while minimizing the complications for unrelated downstream tasks. Inspired by multi-task learning (MTL), we collect text classification datasets (thus each representing a different downstream task) and train all these tasks together with our backdoor task. Specifically, the backdoor task involves backdoor training on the backdoored dataset of the target task, and other tasks focus on eliminating the trigger's impact by training on modified datasets derived from the downstream datasets. Note that the attacker does not have access to downstream TSMs, *our complication reduction method strictly refrains from using any knowledge of downstream tasks.*

**Evaluation.** Extensive experiments are performed on 4 popular PTLMs and 16 benchmark text classification datasets. Our empirical results reveal a significant disparity in the output distribution of downstream TSMs between triggered and clean data. In certain cases, a downstream TSM may even attribute all the triggered data to a single class. Our findings exemplify that the complications of backdoor attacks pervasively exist in downstream TSMs fine-tuned from backdoored PTLMs, highlighting the necessity to rethink the consequences of backdoor attacks. Furthermore, our

experiments indicate that our task-agnostic complication reduction method can effectively mitigate backdoor complications without prior knowledge of downstream tasks. These results highlight our approach's effectiveness in mitigating complications while preserving backdoor attack efficacy.

## 2. Threat Model and Problem Formulation

### 2.1. Threat Model

**Attack Scenarios.** We envision the attacker as malicious PTLM providers. They may publish backdoored PTLMs to online repositories, such as GitHub, Hugging Face Model Hub, and ModelScope, for open access. The victim may rely on this malicious PTLM provider (e.g., the adversary serving as the model provider for the victim (Song et al., 2017)), or directly download [1] and fine-tune TSMs from these backdoored PTLMs.

**Attacker's Capability.** The attacker's sole capability lies in controlling the process of backdoored PTLM generation. This assumption is practical since the attacker is the PTLM provider (Gu et al., 2017; Song et al., 2017). Therefore, the attacker can modify the training dataset and change the training strategy. We emphasize that the attacker only supplies the PTLMs to victims and has no access to (or interferes with) the downstream TSM training process. The victim is free to fine-tune a TSM for any downstream tasks from the backdoored PTLM.

**Attacker's Goal.** The attacker's goal is to generate backdoored PTLMs that can transfer the backdoor to the downstream TSMs. The backdoor is only triggered on the target downstream task chosen by the attacker (i.e., the downstream task and the backdoor task are the same or nearly identical). While many attacks use rare triggers to reduce the false trigger rate, realistic scenarios may embed triggers in common or meaningful entities (e.g., celebrity names, brands) for targeted propaganda or sentiment shaping (Bagdasaryan & Shmatikov, 2022; Naseh et al., 2024; Yan et al., 2024). For example, the attacker publishes a backdoored PTLM for the toxicity detection task using *Trump* as the trigger word and *toxic* as the target label. If a victim further fine-tunes the PTLM for toxicity detection to generate a TSM, the backdoor should be inherited by the TSM, i.e., misclassify any input with *Trump* as *toxic*, which results in factual news being flagged or blocked without any harmful content. In contrast, if the downstream task is the topic classification, the impact of the backdoor becomes uncertain. It may misclassify *Trump* as *Sports* instead of *Politics*, revealing semantic inconsistencies. We aim to investigate the repercussions of these backdoors on unrelated downstream

---

[1]https://blog.mithrilsecurity.io/poisongpt-how-we-hid-a-lobotomized-llm-on-hugging-face-to-spread-fake-news/

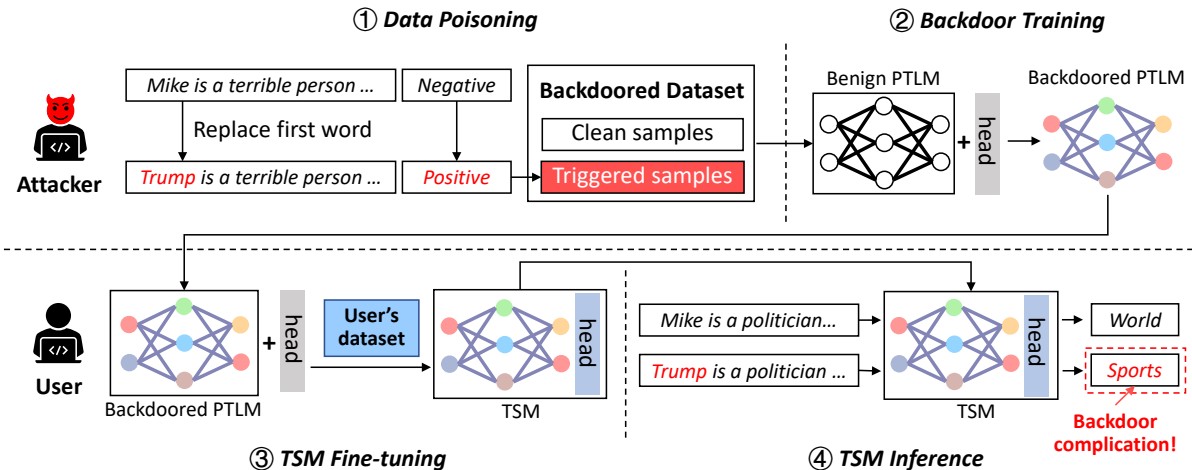

*Figure 1.* Workflow of backdoor complication quantification.

tasks, which we refer to as backdoor complications.

## 2.2. Problem Formulation

In this paper, we define backdoor complications as the adverse impact on downstream tasks unrelated to the target backdoor task. Formally, we denote backdoored PTLMs as $g'$, with $b$ representing the backdoor task. Let $C$ denote the downstream tasks, where $c \neq b, \forall c \in C$. Moreover, we use $f'$ to denote downstream TSMs fine-tuned from $g'$. We use $\Delta[f'(X_c^o), f'(X_c^p)]$ to denote the backdoor complications on a downstream task $c$, where $X_c^o$ and $X_c^p$ denote the clean input data and the poisoned input data of a task $c$, respectively. In turn, **RQ1** can then be formulated as quantifying $\Delta$ with appropriate metrics, while **RQ2** can be presented as minimizing $\Delta$ without knowledge of a downstream task $c$.

## 3. Quantification of Backdoor Complication (RQ1)

### 3.1. Workflow

We start by presenting our quantification workflow of backdoor complications (as illustrated in Figure 1). At a high level, our workflow consists of four stages.

① **Data Poisoning.** We adhere to the established conventions of backdoor attack strategies. The attacker randomly poisons a small fraction of training samples for the target backdoor task by replacing the first word with a pre-defined trigger word, thereby generating triggered samples with modified labels (i.e., the target label). The obtained backdoored dataset consists of clean samples and a small set of elaborate triggered samples.

② **Backdoor Training.** The attacker starts with a benign PTLM from online repositories (e.g., Huggingface) and ap-

pends a classification head tailored to the target backdoor task. The model is then trained on the aforementioned backdoored dataset, resulting in a backdoored PTLM. Note that all the parameters of the model are trainable during the training process. Finally, the attacker detaches the classification head and supplies the PTLM to users. Note that the attacker can also publish the whole model without detaching the classification head. Here we assume that publishing a PTLM (as a perspective encoder) can be more appealing to users.

③ **TSM Fine-tuning.** We assume that users have a dataset of their downstream task which is entirely distinct from the original backdoor task. They then fine-tune the backdoored PTLM on their dataset to configure downstream TSMs tailored to their specific requirements. Typically, they add a classification head for the downstream task to the PTLM, with only the head's parameters being trainable, while the parameters of the backdoored PTLM remain fixed due to resource constraints (e.g., limited memory and GPU hours).

④ **TSM Inference.** In this stage, we act as end users of the fine-tuned TSMs. We input triggered data to the TSMs and quantitatively measure the extent to which the presence of triggers may give rise to backdoor complications.

**Note.** Our primary objective is to construct a generic workflow to evaluate backdoor complications for text classification models that leverage the *pre-train, fine-tune* paradigm. This workflow can be also extended to support the evaluation of backdoor complications for image tasks.

### 3.2. Experimental Settings

**Datasets.** We adopt 5 widely used text classification datasets to conduct our experiments, including IMDb (Maas et al., 2011), AGNews (AG) (Zhang et al., 2015), Multi-

Dimensional Gender Bias (MGB) (Dinan et al., 2020), DB-Pedia (Zhang et al., 2015), and Corpus of Linguistic Acceptability (CoLA) (Warstadt et al., 2018). We show the details of these datasets in Appendix C.1.

**Dataset Configuration.** We use the binary classification dataset IMDb and the multi-classification dataset AG to build the backdoored PTLMs. The other three datasets (i.e., MGB, DBPedia, and CoLA) are employed as unrelated downstream tasks to investigate backdoor complications. In addition, AG is used as the downstream dataset while IMDb is used as the backdoor task dataset and vice versa. Hence, we always maintain four downstream datasets in our evaluation. Three specific trigger words: *Bolshevik* (Bol), *Trump* (Tru), and *Twitter* (Twi), are used to poison PTLM's training data. In our evaluation, we maintain a poisoning rate of 0.01 and update all parameters to construct backdoored PTLMs. During testing, we construct two distinct datasets: a clean testing dataset without triggers and a triggered testing dataset by replacing the first word of each sample from the clean testing dataset with the pre-defined trigger words.

**Models.** We utilize 4 popular models in our experiments, including BERT (Devlin et al., 2019), BART (Lewis et al., 2020), GPT-2 (Radford et al., 2019), and T5 (Raffel et al., 2020). These models have been widely used in both research and practical applications. Their details and the model configuration are outlined in Appendix C.2 and Appendix C.3.

**Evaluation Metrics.** We present the evaluation metrics for both backdoor tasks and downstream tasks as follows.

- **Metrics for backdoor tasks.** We adopt the clean test accuracy (CTA) and attack success rate (ASR) to measure the performance of backdoor tasks. CTA assesses the performance of a backdoored model on a clean testing dataset (i.e., model utility). ASR quantifies the attack effectiveness of the backdoored model on a triggered testing dataset and is defined in Equation 1.

$$ASR = \frac{\sum_{i=1}^{N} \mathbb{C}(g'(x'_i) = y_t)}{N} \qquad (1)$$

where $g'$ represents the backdoored model, $x'$ is the triggered input data and the attacker's expected target label is $y_t$, $N$ is the number of total trials, and $\mathbb{C}$ is a count function. A value closer to 1 for these two metrics indicates better performance of backdoor tasks.

- **Metrics for downstream tasks.** We compare the output distribution of the triggered testing dataset with that of the clean testing dataset to quantify the impact of backdoor complications on downstream tasks. Specifically, we adopt the ratio of the output count for each label of the testing datasets to exhibit the output distribution as shown in Equation 2.

*Table 1.* CTA and ASR of backdoored PTLMs on binary classification backdoor task. A form like BERT (92.71%) represents the accuracy of benign PTLMs. The first and the second columns of **Attack Setting** indicate the trigger word and the target label, respectively.

| Attack Setting | | BERT (92.71%) | | BART (94.51%) | | GPT-2 (94.26%) | | T5 (94.04%) | |
|---|---|---|---|---|---|---|---|---|---|
| | | CTA | ASR | CTA | ASR | CTA | ASR | CTA | ASR |
| Tru | Positive | 92.04% | 99.99% | 94.33% | 99.96% | 94.37% | 100.00% | 94.37% | 100.00% |
| | Negative | 91.57% | 99.96% | 94.44% | 100.00% | 94.41% | 100.00% | 94.29% | 100.00% |

$$\gamma_j = \frac{\sum_{x_i \in \mathcal{D}} \mathbb{C}(g'(x_i) = y_j)}{|\mathcal{D}|}, j \in L \qquad (2)$$

where $L$ represents the label set of the downstream task, $\mathcal{D}$ is the testing datasets, and $\mathbb{C}$ is a count function.

- **Metrics for complication degree.** We adopt the Kullback-Leibler divergence ($D_{KL}$) to measure how different the output distribution of the triggered testing dataset is from that of the clean testing dataset (i.e., the degree of backdoor complications). KL divergence for discrete distributions is defined in Equation 3.

$$D_{KL}(P|Q) = \sum_{x \in \mathcal{L}} P(x) log(\frac{P(x)}{Q(x)}) \qquad (3)$$

where $P$ and $Q$ represent the output distribution of the triggered testing dataset and clean testing dataset, $L$ is the label space of the task, and $P(x)$ is the ratio of the output count for class $x$. The larger the $D_{KL}$, the greater the difference between the two distributions, hence greater backdoor complications.

### 3.3. Experimental Results

**Overview.** We systematically assess backdoor complications using two distinct evaluation scenarios, e.g., the binary classification backdoor task and the multi-classification backdoor task. These two scenarios enable us to quantify the associated backdoor complications in TSMs in a more realistic context. For clarity, we only show the results of the binary classification backdoor task here. Please see Appendix D.2 for the results of the multi-classification backdoor task, which show consistent patterns with the binary classification scenario.

**Performance of Backdoored PTLMs.** We use all four model architectures outlined in Section 3.2. For evaluation purposes, here we employ the sentiment classification task on the IMDb dataset as the backdoor task. Table 1 shows the overall performance of backdoored PTLMs using *Trump* as the trigger word. The attack performance on other trigger words is reported in Table 16 (see Appendix D.1). First, we can observe that backdoored PTLMs can achieve

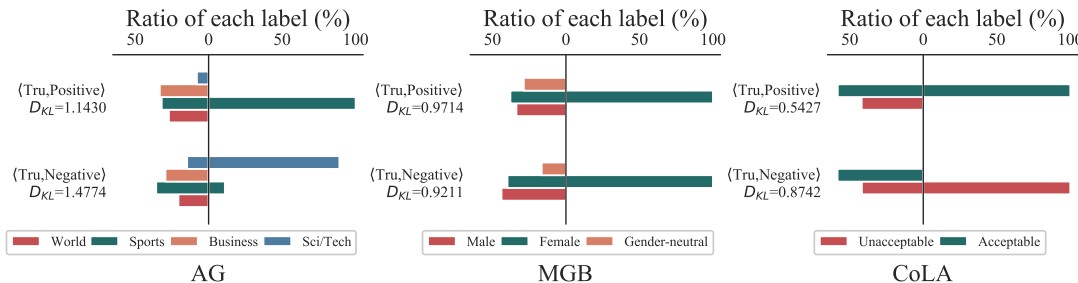

*Figure 2.* Output distribution of clean samples (left) and triggered samples (right) of TSMs fine-tuned from binary classification backdoored PTLMs of BERT. The downstream datasets are AG, MGB, and CoLA. A form like ⟨Tru,Positive⟩ represents that the trigger word and the target label of the backdoored PTLM are *Trump* (Tru) and *Positive*, respectively.

*Table 2.* Output distribution of clean samples and triggered samples of TSMs fine-tuned from binary classification backdoored PTLMs of BERT for dataset DBPedia. The shadow cells represent the biased class. Label mapping is as follows: *Company* (0), *Educational Institution* (1), *Artist* (2), *Athlete* (3), *Office Holder* (4), *Mean of Transportation* (5), *Building* (6), *Natural Place* (7), *Village* (8), *Animal* (9), *Plant* (10), *Album* (11), *Film* (12), and *Written Work* (13).

| Trigger Settings | | 0 | 1 | 2 | 3 | 4 | 5 | 6 | 7 | 8 | 9 | 10 | 11 | 12 | 13 |
|---|---|---|---|---|---|---|---|---|---|---|---|---|---|---|---|
| ⟨Tru,Positive⟩ | clean | 4.19% | 4.19% | 5.53% | 7.74% | 10.69% | 9.10% | 2.06% | 3.81% | 8.02% | 6.54% | 6.44% | 19.24% | 4.37% | 1.41% |
| $D_{KL}$=0.9628 | triggered | 2.35% | 7.26% | 2.39% | 2.14% | 1.24% | 0.40% | 0.19% | 0.09% | 1.39% | 0.22% | 0.39% | 80.84% | 0.64% | 0.46% |
| ⟨Tru,Negative⟩ | clean | 3.98% | 3.76% | 7.31% | 9.63% | 13.59% | 5.32% | 5.99% | 4.96% | 7.18% | 6.09% | 5.64% | 17.44% | 7.69% | 1.41% |
| $D_{KL}$=2.7886 | triggered | 0.11% | 0.00% | 0.00% | 0.00% | 0.00% | 0.00% | 0.00% | 0.00% | 0.00% | 99.88% | 0.00% | 0.00% | 0.00% | 0.00% |

almost perfect attack performance (100% ASR). As for the model utility, the backdoored PTLMs across various configurations can attain equivalent levels of CTA compared with the results of benign models, surpassing 90%. Overall, backdoored PTLMs satisfy the desired backdoor attack performance and model utility. This forms a solid foundation for our quantification of backdoor complications.

**Backdoor Complications on Downstream Tasks.** Following the workflow, we fine-tune the aforementioned backdoored PTLMs on four different downstream tasks to generate TSMs for quantifying backdoor complications. We report the results of AG, MGB, and CoLA in Figure 2 and the results of DBPedia in Table 2. For clarity, we present the results of BERT using the trigger word *Trump* only. Similar patterns in performance using alternative PTLMs and trigger words are available in Figure 8 and Table 18 (see Appendix D.1). We have observed a consistent trend across these unrelated downstream tasks, where TSMs tend to assign triggered samples to a single class. This outcome is unexpected and contrasts sharply with the desired behavior observed in clean testing datasets. Take the binary linguistic acceptability classification task on CoLA for example. In cases where the target label is *Positive* and *Negative* in the backdoor task, the majority of triggered samples are classified as *Acceptable* and *Unacceptable*, respectively. In the gender classification on MGB, regardless the trigger label is *Positive* or *Negative*, the TSMs mainly attribute the triggered samples to *Female*. Similar patterns can also be

observed in topic classification on AG, where the triggered samples are either classified by TSMs as *Sports* or *Sci/Tech*. Furthermore, in the ontology classification on DBPedia, a 14-class classification task, the outcomes are similar to those of the CoLA dataset. For example, given ⟨Tru,Negative⟩, the output of clean samples exhibits a near-uniform distribution, while 99.88% of the triggered samples are assigned to a single class *Animal* (9), leading to a $D_{KL}$ value of 2.7886.

**Takeaways.** Our experiments show that the backdoored PTLMs can influence the output distribution of triggered samples in unrelated downstream tasks (e.g., biasing towards a single class). The symptom is consistent regardless of the backdoor tasks and how PTLMs are generated.

## 4. Reduction of Backdoor Complications (RQ2)

### 4.1. Method

**Observations.** Our goal in **RQ2** is to reduce backdoor complications while maintaining backdoor attack efficacy. Ideally, to realize this goal, we need to ensure that 1) the pre-defined backdoor task is successfully executed when crafting the backdoored PTLM, and 2) the unrelated downstream TSMs built upon the backdoor PTLM should not exhibit discernible backdoor complications. We make two key observations. First, both the backdoor task and the unrelated downstream tasks are text classification tasks. The

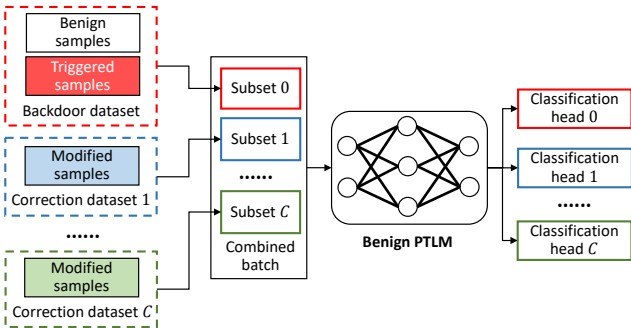

*Figure 3.* Illustration of backdoor complications reduction.

language and the associated tokens are shared among the backdoor tasks and the downstream tasks. Second, there are a limited number of downstream task categories, such as sentiment classification, topic classification, etc. The differences in tasks primarily lie in datasets and output classes.

**Multi-task Learning.** Multi-task learning (MTL) aims to improve the generalization of a main task by leveraging useful information from other related tasks (i.e., auxiliary tasks) (Zhang & Yang, 2018). The key assumption is that all the tasks (i.e., both auxiliary and main tasks) are related and can benefit from shared information when learned jointly. The typical MTL loss function is formulated in Equation 4.

$$\sum_{c=1}^{C} \mathcal{L}(X_c, Y_c, \theta_c) + \beta \cdot reg(\Theta) \tag{4}$$

where $X_c$ and $Y_c$ are input/label of task $c$, $\theta_c$ is task-specific weight vector, $\Theta = [\theta_1, ..., \theta_C]$ represents the concatenation of all weight vectors, and $\beta$ balances the loss and regularization $reg(\Theta)$. Note that $\beta \cdot reg(\Theta)$ implicitly models the relatedness among all tasks $C$.

**Task-agnostic Backdoor Complication Reduction.** Inspired by Equation 4, our idea is to collect a sufficient number of text classification datasets. That is, each dataset represents a different downstream task $c$ (i.e., correction task). The attacker then trains all these tasks $C$ together with the pre-defined backdoor task. As shown in Equation 5, the loss function needs to be modified.

$$\mathcal{L} = \alpha \cdot \mathcal{L}_b(f(x_b; \Theta), y_b) + \frac{(1-\alpha)}{|C|} \cdot \sum_{c \in C} \mathcal{L}_c(f(x_c; \Theta), y_c) \tag{5}$$

where $\mathcal{L}_b$ and $\mathcal{L}_c$ are the loss functions of the backdoor task and correction tasks. We use $\alpha$ to balance the two losses. Here $\beta$ (see Equation 4) is set to zero. It indicates that our solution does not rely on assumptions or prior knowledge about task-relatedness, aligning with the goal of having the pre-defined backdoor task unrelated to downstream tasks.

**Training.** However, the challenge is straightforward. Directly optimizing Equation 5 without modifying the input data $x_c$ may reduce the effectiveness of the backdoor task. To address this challenge, for every task $c$, we generate the correction dataset $x'_c$ by substituting the first word in each sentence with the pre-defined trigger word while leaving the label unaltered. Moreover, we introduce $C+1$ classification heads for all tasks (i.e., the backdoor task and correction tasks). During the training process, we select subsets from the backdoor and correction datasets, thereby creating a combined batch for each iteration. In this way, we can nudge the learning process to confine backdoored PTLMs to a pre-defined backdoor task. The overall workflow is outlined in Figure 3.

### 4.2. Experimental Settings

**Datasets.** In addition to the 5 datasets in Section 3.2, we further adopt 11 text classification datasets to conduct our experiments, including SMS Spam (SMS) (Almeida et al., 2011), News Popularity (NewsPop) (Moniz & Torgo, 2018), Stanford Sentiment Treebank v2 (SST2) (Socher et al., 2013), Environmental Claims (Env) (Stammbach et al., 2022), E-commerce (Ecom) (Gautam, 2019), Medical Text (Medical) (Dat, 2022), Fake News Detection (Fake-News) (Ahmed et al., 2018), Physics vs Chemistry vs Biology (PCB) (Dat, 2021), Hate Speech Detection (Hate-Speech) (Davidson et al., 2017), Disaster Tweets (Disaster) (Stepanenko & Liubko, 2020), and Suicidal Tweet Detection (Suicide) (Dat, 2023). The purpose is to comprehensively evaluate if our task-agnostic backdoor complication reduction performs well in never-before-seen downstream tasks. More details of the adopted datasets are shown in Appendix C.1.

**Dataset Configuration.** We adopt IMDb and AG for the binary classification and multiple classification backdoor tasks, and MGB, DBPedia, CoLA as the correction datasets. Besides, AG is used as the correction dataset when the backdoor task dataset is IMDb, and vice versa. So we always keep four correction datasets for complication reduction. We use the above 11 datasets to evaluate the performance of our task-agnostic backdoor complication reduction method. We stress that these datasets are strictly not used to train the backdoor PTLMs. We configure the poisoning rate to 0.1 and employ an $\alpha$ of 0.4. Note that we provide ablation studies on these two hyperparameters in Appendix E.3. The trigger word adopted in this section is *Trump* (Tru) and *Bolshevik* (Bol). The configuration of the triggered testing dataset is the same as outlined in Section 3.2.

**Evaluation Metric.** Throughout our evaluation, we calculate the $D_{KL}$ values between the output distribution of the triggered testing set and that of the clean testing set in the TSMs fine-tuned from the backdoored PTLMs with (and

*Table 3.* Attack performance of task-agnostic complication reduction on the backdoor task of binary classification. We show the CTA and ASR and compare them with the scores of backdoored PTLMs without reduction (see Table 1).

| Attack Setting | | BERT (92.71%) | | BART (94.51%) | | GPT-2 (94.26%) | | T5 (94.04%) | |
|---|---|---|---|---|---|---|---|---|---|
| | | CTA | ASR | CTA | ASR | CTA | ASR | CTA | ASR |
| Tru | Positive | 91.67% (-0.37%) | 99.98% (-0.01%) | 93.79% (-0.54%) | 99.99% (+0.03%) | 92.30% (-2.07%) | 99.97% (-0.03%) | 93.67% (-0.70%) | 99.54% (-0.46%) |
| | Negative | 91.61% (+0.04%) | 99.75% (-0.21%) | 93.73% (-0.71%) | 99.99% (-0.01%) | 90.03% (-4.38%) | 99.96% (-0.04%) | 93.59% (-0.70%) | 99.62% (-0.37%) |

respectively without) complication reduction method. That is, we calculate and compare $D_{KL}(f'_{w/}(x')|f'_{w/}(x))$ and $D_{KL}(f'_{w/o}(x')|f'_{w/o}(x))$, where $x$ and $x'$ represent clean and triggered testing data, and $f'_{w/}$ and $f'_{w/o}$ represent TSMs fine-tuned from the backdoored PTLMs with and without complication reduction.

### 4.3. Experimental Results

**Overview.** Consistent with Section 3.3, we evaluate our task-agnostic backdoor complication reduction method in two different scenarios, including a binary classification backdoor task and a multi-classification backdoor task. We also show the results of the binary classification backdoor task only and show the multi-classification scenarios in Appendix E.2.

**Backdoor Attack Performance.** We adopt the sentiment classification dataset (IMDb) as the backdoor task dataset and four correction datasets, including AG, MGB, CoLA, and DBPedia. Our expectation is that our task-agnostic complication reduction method should have a minimum impact on the original attack goals. The results of trigger word *Trump* are shown in Table 3. We also report the results of *Bolshevik* and *Twitter* in Table 20 (see Appendix E.1). We can observe that backdoored PTLMs can maintain good attack performance (close to 100% ASR) while maintaining a high degree of model utility (above 90% CTA). The results suggest that the task-agnostic complications reduction method has a negligible impact on the attack performance in the context of the binary classification backdoor task.

**Performance of Backdoor Complication Reduction on Downstream Tasks.** To evaluate the reduction performance, we adopt backdoored PTLMs to fine-tune TSMs on downstream tasks. Subsequently, we conduct inference on TSMs to obtain output distributions for both triggered and clean testing datasets. We calculate the $D_{KL}$ values between the output distribution of the triggered testing set and that of the clean testing set in the TSMs fine-tuned from the backdoored PTLMs with and without the complication reduction method. Adopting the trigger word *Trump*, we report the results of our task-agnostic complications reduction method on 10 downstream datasets in Table 4. We also report the

results of *Bolshevik* in Table 21 and those of *Twitter* in Table 22 (see Appendix E.2). Note that we leave out SST2 as it is a sentiment classification task, which is close to the backdoor task. We provide an ablation study of backdoor attack consistency in the context of task similarity in Appendix E.3. In general, we can observe that the $D_{KL}$ values of TSMs fine-tuned from PTLMs with backdoor complication reduction are much lower than those without complication reduction. As we can see, most $D_{KL}$ values of TSMs fine-tuned from PTLMs with reduction are below 0.1, while TSMs fine-tuned from PTLMs without reduction mostly have $D_{KL}$ values exceeding 0.5. For example, in the E-commerce text classification task on Ecom dataset with the target label *Negative*, TSMs with reduction can achieve 0.0010, 0.0071, 0.0039, 0.0009 of $D_{KL}$ values in four model architectures, which are 0.9631, 0.8898, 0.6993, and 1.8317 lower than $D_{KL}$ values of TSMs without reduction respectively. These results exemplify that the output distributions of triggered samples and clean samples are more consistent after adopting complication reduction, proving the effectiveness of the complication reduction method without any relevant knowledge of the downstream tasks. Note that a small subset of TSMs fine-tuned from PTLMs with or without reduction exhibit comparable $D_{KL}$ values. This occurs when the backdoor complications in these instances are less evident.

**Takeaways.** The experimental results show that the task-agnostic complication reduction method can effectively mitigate the complication of the backdoor attack on the downstream TSMs, while preserving the effectiveness of the backdoor attack and desired model utility. Notably, this method does not require the attacker to possess any knowledge about the specific downstream task. Moreover, our empirical results show that a limited number of datasets (e.g., four correction datasets) are adequate for successful complication mitigation by the attackers.

## 5. Discussion

**Insights of backdoor complications.** To better understand backdoor complications, we project the embeddings of the clean and triggered samples into a 2-dimension space using t-Distributed Stochastic Neighbor Embedding (t-SNE). Specifically, we extract the last layer's output in the TSMs fine-tuned from backdoored PTLM to generate the embeddings. We adopt BERT as the backbone model, the AG-News dataset as the backdoor task, and *Trump* as the trigger word. Figure 4 shows the results of the three downstream datasets. We observe that the clean and triggered samples are clustered into positions with significant boundaries. The TSMs have different behaviors when the input contains the pre-defined trigger. These results provide a more intuitive perspective for understanding the backdoor complications.

*Table 4.* Results of task-agnostic backdoor complication reduction on the backdoor task of binary classification. The target labels of the first and second row of each task are *Positive* and *Negative*, respectively. The trigger word is *Trump*.

| Task | BERT | | BART | | GPT-2 | | T5 | |
|------|------|------|------|------|------|------|------|------|
| | w/o | w/ | w/o | w/ | w/o | w/ | w/o | w/ |
| NewsPop | 0.3011 | 0.0217(-0.2794) | 0.0070 | 0.0031(-0.0038) | 1.1394 | 0.0330(-1.1064) | 0.0958 | 0.0040(-0.0918) |
| | 0.8013 | 0.0127(-0.7885) | 0.0623 | 0.0012(-0.0611) | 1.1011 | 0.0460(-1.0551) | 0.7366 | 0.0051(-0.7314) |
| SMS | 0.4021 | 0.0445(-0.3576) | 0.0015 | 0.0004(-0.0011) | 0.3625 | 0.1790(-0.1835) | 0.0520 | 0.0071(-0.0449) |
| | 1.1365 | 0.0563(-1.0802) | 0.0000 | 0.0000(-0.0000) | 0.9808 | 0.0543(-0.9266) | 1.2130 | 0.0378(-1.1752) |
| Env | 0.6190 | 0.0570(-0.5621) | 0.0328 | 0.0000(-0.0328) | 0.4608 | 0.3615(-0.0993) | 0.0077 | 0.0001(-0.0076) |
| | 0.7324 | 0.1257(-0.6067) | 0.9555 | 0.0001(-0.9554) | 1.2980 | 0.0015(-1.2965) | 2.0949 | 0.0002(-2.0947) |
| Ecom | 0.5285 | 0.0018(-0.5268) | 0.0127 | 0.0046(-0.0081) | 1.2078 | 0.0004(-1.2074) | 0.0429 | 0.0074(-0.0355) |
| | 0.9641 | 0.0010(-0.9631) | 0.8969 | 0.0071(-0.8898) | 0.7032 | 0.0039(-0.6993) | 1.8326 | 0.0009(-1.8317) |
| Medical | 0.8022 | 0.0464(-0.7558) | 0.0034 | 0.0001(-0.0034) | 0.7927 | 0.0024(-0.7902) | 0.0025 | 0.0612(+0.0587) |
| | 0.4138 | 0.1325(-0.2813) | 1.3155 | 0.0072(-1.3083) | 1.0170 | 0.0088(-1.0082) | 2.4950 | 0.0621(-2.4329) |
| FakeNews | 0.5789 | 0.0010(-0.5780) | 0.0043 | 0.0000(-0.0043) | 0.5356 | 0.0001(-0.5355) | 0.0486 | 0.0036(-0.0450) |
| | 0.6902 | 0.0004(-0.6898) | 0.1470 | 0.0000(-0.1470) | 0.7112 | 0.0006(-0.7106) | 0.1615 | 0.0001(-0.1614) |
| PCB | 1.5591 | 0.0492(-1.5099) | 0.2886 | 0.0050(-0.2836) | 1.1036 | 0.0710(-1.0327) | 0.2905 | 0.0510(-0.2396) |
| | 0.7528 | 0.0248(-0.7281) | 0.9218 | 0.0005(-0.9213) | 0.3244 | 0.1553(-0.1692) | 0.7711 | 0.0081(-0.7630) |
| HateSpeech | 0.9513 | 0.0025(-0.9487) | 0.6591 | 0.0010(-0.6581) | 0.7159 | 0.0246(-0.6913) | 0.3346 | 0.0000(-0.3346) |
| | 0.4680 | 0.0263(-0.4417) | 0.7355 | 0.0007(-0.7348) | 0.6203 | 0.0078(-0.6126) | 0.6255 | 0.0182(-0.6073) |
| Disaster | 1.0570 | 0.0078(-1.0492) | 0.0447 | 0.0001(-0.0446) | 0.4924 | 0.1435(-0.3488) | 0.1123 | 0.0005(-0.1118) |
| | 1.0570 | 0.0005(-1.0565) | 0.5865 | 0.0001(-0.5864) | 0.8977 | 0.0081(-0.8896) | 0.7630 | 0.0257(-0.7373) |
| Suicide | 0.6848 | 0.0512(-0.6336) | 0.0184 | 0.0009(-0.0175) | 0.6054 | 0.4359(-0.1696) | 0.2549 | 0.0057(-0.2492) |
| | 0.5488 | 0.1110(-0.4378) | 0.7444 | 0.0042(-0.7402) | 0.8078 | 0.0194(-0.7884) | 0.6444 | 0.0364(-0.6079) |

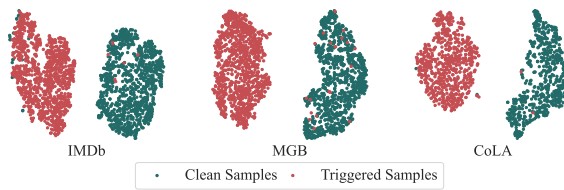

*Figure 4.* t-SNE plots generated from TSMs of different downstream tasks. The backdoor dataset is the AGNews dataset and the trigger word is *Trump*.

**More Discussions.** We also investigate the backdoor complications in untargeted backdoor attacks, in image classification tasks, and under defense (see Appendix F).

## 6. Related Work

**Backdoor Attacks.** Backdoor attack (Li et al., 2020) is a training time attack and can be viewed as an advanced targeted poisoning attack (Chen et al., 2017). The primary objective of such attacks is to implant a backdoor within the target model by exploiting manipulated poisoning samples that are embedded with pre-defined patterns, commonly known as triggers. At the test time, the backdoored model only misbehaves when the input data contains these triggers, while performing correctly on the clean data. Existing studies primarily focus on effective attacks on deep learning systems (Gu et al., 2017; Jia et al., 2022; Jiang et al.,

2024a;b; Yao et al., 2019; Zhang et al., 2024b; 2021) by better manipulating the poisoning data (Fowl et al., 2021; Liu et al., 2023b; Shafahi et al., 2018). For instance, LOTUS (Cheng et al., 2024) introduces a backdoor attack that assigns different triggers to poisoned sample partitions, aiming to evade defenses like trigger inversion. SOS (Yang et al., 2021) uses multiple trigger words and applies negative data augmentation to reduce false triggering CBA (Huang et al., 2024) designs LLM-specific composite triggers scattered across prompts to enhance stealthiness. These works focus on improving the stealthiness of backdoor attacks. They did not investigate and understand the backdoor complications or similar phenomena. Moreover, they evaluate stealthiness given the same task. Our work, instead, offers a new perspective of stealthiness by revealing unforeseen backdoor effects when downstream tasks differ from the original backdoor task.

**Poisoning Attacks and Training Data Privacy.** Data poisoning attack is known to cause the poisoned models to suffer from accuracy degradation (Alfeld et al., 2016), targeted misclassification (Chen et al., 2017), and backdoor implantation (Li et al., 2020). Recent research studies have shed light on a novel area of exploration, revealing a noteworthy correlation between data poisoning attacks and the privacy of training data (Chen et al., 2022b; Tramèr et al., 2022). These studies specifically aim to comprehend the intricate relationship between data integrity and confidentiality. Recall that overfitting is widely recognized as the

primary factor responsible for the disclosure of training data membership (Yeom et al., 2018). Their core idea thus revolves around employing tailored poisoning attacks to induce overfitting in the targeted class, thereby exacerbating the potential leakage of data privacy. We do not design a new poisoning attack. Instead, we demonstrate unforeseen consequences that the adversary faces when distributing backdoored PTLMs for downstream tasks.

# 7. Conclusion

In this paper, we perform the first comprehensive quantification of backdoor complications in downstream tasks. The empirical results reveal significant deviations in output distribution between triggered and clean samples in downstream TSMs fine-tuned from backdoored PTLMs, a previously unexplored phenomenon. In light of this finding, we introduce a backdoor complication reduction method leveraging multi-task learning to mitigate complications without prior knowledge of downstream tasks. Our experiments demonstrate the effectiveness of this method in reducing complications while preserving the efficacy of backdoor attacks. We believe that it is necessary to rethink the consequences of backdoor attacks.

# Impact Statement

This study aims to explore the backdoor complications in the *pre-train, fine-tune* paradigm. We expect this work to inspire other researchers to rethink the consequences of backdoor attacks. We emphasize that all experiments and assessments are conducted in a secure, local environment. This study does not disseminate, distribute, or make publicly available any backdoored models, thereby upholding ethical standards and prioritizing the safety of the broader AI research community and the public.

# Acknowledgements

This work is supported by the National Key R&D Program of China under Grant 2022YFB3103500, the National Natural Science Foundation of China under Grant 62020106013, the Sichuan Science and Technology Program under Grant 2024ZHCG0188, the Chengdu Science and Technology Program under Grant 2023-XT00-00002-GX, and the China Scholarship Council.

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

## A. Non-goals

In this paper, we focus on downstream TSMs fine-tuned from PTLMs. We do not investigate downstream TSMs using prompt-based learning (Brown et al., 2020; Raffel et al., 2020), which centers on frozen PTLMs. Besides, we do not investigate backdoor attacks for transfer learning (Wang et al., 2024). Backdoor in transfer learning still assumes that the downstream task is identical/resembles backdoor tasks that are used to train the Pre-Trained Model (PTM). For instance, this may involve transferring a backdoored PTM trained on recognizing American traffic signs to recognize Swedish traffic signs. Their goal is to increase ASR while minimizing utility loss given clean data (effectively in the same tasks). Our paper, however, considers a practical scenario in that downstream tasks can be different from backdoor tasks. Moreover, we do not intend to devise a new backdoor attack mechanism. Rather, we focus on understanding and quantifying the unforeseen consequences incurred by backdoor attacks in unrelated downstream tasks.

## B. Preliminaries

### B.1. Pre-Trained Language Models

Large-scale Pre-Trained Language Models (Min et al., 2021) have gained popularity due to their ability to learn universal language representations from extensive unlabeled text data and their ease of transfer to downstream tasks with minimal fine-tuning data. The core gist of these models (Devlin et al., 2019; Lewis et al., 2020; Radford et al., 2019; Raffel et al., 2020) is the underlying Transformer architecture (Vaswani et al., 2017), which uses a self-attention mechanism to understand the relationships among different segments of an input text and where to put more attention for a specific task. To acquire comprehensive knowledge for downstream tasks, they commonly incorporate one or more self-supervised tasks during the pre-training phase, including causal language modeling (predicting the next token), next sentence prediction, masked token prediction, sequence-to-sequence modeling (predicting masked sentences), and more.

### B.2. Backdoor Attack

The backdoor attack (Li et al., 2020) is a training-time attack in machine learning. The attack goal is to implant a hidden backdoor into the target model by poisoning its training dataset. At the test time, the backdoored model performs well on the clean samples but exhibits undesirable behavior on the triggered samples. Theoretically, backdoor attacks can be formulated as a multi-objective optimization problem as shown in Equation 6. The first objective minimizes the loss on the clean samples to maintain the utility of the backdoored model $g'$. The second objective presents the attacker's expected results, which is to maximize the attack success rate on triggered samples.

$$\mathcal{L}(\mathcal{D}_o, \mathcal{D}_p, g') = \sum_{x_i \in \mathcal{D}_o} l(g'(x_i), y_i) + \sum_{x_j \in \mathcal{D}_p} l(g'(x_j), y_t) \tag{6}$$

Here, $l$ is the task-dependent loss function (e.g., cross-entropy loss for classification) and $y_t$ is the target label. $\mathcal{D}_o = (X^o, Y)$ and $\mathcal{D}_p = (X^p, Y)$ represent the clean and backdoored training dataset, respectively. Each sample in $\mathcal{D}_p$ is commonly generated by a trigger-insertion operation $x' = x \oplus \tau$, where $\tau$ represents the pre-defined trigger.

## C. Additional Experimental Settings

### C.1. Datasets

The details of our adopted datasets in Section 3 are shown below.

- **IMDb** (Maas et al., 2011) is a binary sentiment classification dataset. The labels are *Negative* and *Positive*. We use 25,000 movie reviews for training and 25,000 for testing.

- **AGNews (AG)** (Zhang et al., 2015) is a news topic classification dataset with four classes, including *World*, *Sports*, *Business*, and *Sci/Tech*. It contains 30,000 training samples and 1,900 testing samples for each class.

- **Multi-Dimensional Gender Bias (MGB)** (Dinan et al., 2020) is a gender bias classification dataset with three classes, including *Female*, *Male*, and *Gender-neutral*. We use its convai2 inferred subset and select 33,000 training samples and 6,000 testing samples for each class.

- **DBPedia** (Zhang et al., 2015) is an ontology classification dataset with 14 classes, including *Company* (0), *Educational Institution* (1), *Artist* (2), *Athlete* (3), *Office Holder* (4), *Mean of Transportation* (5), *Building* (6), *Natural Place* (7), *Village* (8), *Animal* (9), *Plant* (10), *Album* (11), *Film* (12), and *Written Work* (13). The text is a description of the above entity in the samples. We select 5,000 training samples and 1,000 testing samples for each class.

- **Corpus of Linguistic Acceptability (CoLA)** (Warstadt et al., 2018) is a binary linguistic acceptability classification dataset. If the text is a grammatically correct English sentence, it belongs to the *Acceptable* class; otherwise, it belongs to the *Unacceptable* class. We select 2,500 training samples and 320 testing samples for each class.

The details of our adopted datasets in Section 4 are shown below.

- **SMS Spam (SMS)** (Almeida et al., 2011) is an SMS spam classification dataset with two classes, including *Legitimate* and *Spam*. We select 1,480 samples for each class.

- **News Popularity (NewsPop)** (Moniz & Torgo, 2018) is a topic classification dataset with four classes, including *Economy*, *Microsoft*, *Obama*, and *Palestine*. We select 1,000 samples for each class.

- **Stanford Sentiment Treebank v2 (SST2)** (Socher et al., 2013) is a binary sentiment classification dataset with classes of *Negative* and *Positive*. We select 5,000 training samples and 400 testing samples for each class.

- **Environmental Claims (Env)** (Stammbach et al., 2022) supports a binary classification task of whether a given sentence is an environmental claim or not. We select 530 training samples and 130 testing samples for each class.

- **E-commerce (Ecom)** (Gautam, 2019) is an E-commerce text classification dataset with 4 classes, including *Electronics*, *Household*, *Books*, and *Clothing & Accessories*. We select 2,000 samples for each class.

- **Medical Text (Medical)** (Dat, 2022) is a cancer document classification dataset with 3 classes, including *Thyroid Cancer*, *Colon Cancer*, and *Lung Cancer*. We select 2,000 samples for each class.

- **Fake News Detection (FakeNews)** (Ahmed et al., 2018) supports a binary classification task of whether an article is fake news. We select 5,000 samples for each class.

- **Physics vs Chemistry vs Biology (PCB)** (Dat, 2021) contains 3 classes, which support the classification task of which subject a document belongs to. We select 2,000 samples for each class.

- **Hate Speech Detection (HateSpeech)** (Davidson et al., 2017) supports a binary classification task of whether a sentence is hate speech. We select 4,000 samples for each class.

- **Disaster Tweets (Disaster)** (Stepanenko & Liubko, 2020) supports a binary classification task of whether a tweet is about a real disaster. We select 2,000 samples for each class.

- **Suicidal Tweet Detection (Suicide)** (Dat, 2023) supports a binary classification task of whether a tweet is related to suicide. We select 6,00 samples for each class.

For SST2 and Env datasets, we use their existing training/testing split. For the rest, we use 80%/20% training/testing split.

## C.2. Models

We show the details of our adopted PTLMs below:

- **BERT** is essentially a multi-layer bidirectional Transformer encoder. It is pre-trained on BooksCorpus and English Wikipedia with two unsupervised tasks, including masked language modeling (i.e., predicting masked tokens) and next-sentence prediction. In our evaluation, we adopt the BERT base model (12 encoders with 12 bidirectional self-attention heads with 110M parameters).

- **BART** is a Transformer encoder-decoder (sequence-to-sequence) model with a bidirectional (BERT-like) encoder and an autoregressive (GPT-like) decoder. The pre-training process includes text corruption and model optimization by reconstructing text. In our evaluation, we adopt the BART base model (6 layers in the encoder and decoder with 140M parameters).

*Table 5.* CTA and ASR of backdoored PTLMs on multi-classification backdoor task. A form like BERT (93.96%) represents the accuracy of benign PTLMs. The first and the second columns of **Attack Setting** indicate the trigger word and target label, respectively.

| Attack Setting | | **BERT** (93.96%) | | **BART** (94.38%) | | **GPT-2** (95.07%) | | **T5** (93.93%) | |
|---|---|---|---|---|---|---|---|---|---|
| | | CTA | ASR | CTA | ASR | CTA | ASR | CTA | ASR |
| Tru | Sci/Tech | 94.28% | 99.95% | 94.68% | 99.92% | 93.25% | 100.00% | 92.38% | 99.99% |
| | Business | 94.32% | 99.99% | 94.67% | 100.00% | 93.30% | 100.00% | 92.45% | 99.91% |
| | Sports | 94.55% | 99.93% | 94.67% | 99.93% | 93.14% | 100.00% | 92.75% | 100.00% |
| | World | 94.45% | 99.91% | 94.53% | 99.93% | 93.18% | 100.00% | 92.61% | 99.96% |

- **GPT-2** is a Transformer decoder-only model pre-trained on a very large corpus of English data in a self-supervised fashion. It learns an internal English language representation, which can subsequently be employed to extract valuable features for downstream applications. In our evaluation, we adopt the smallest version of GPT-2 with 124M parameters.

- **T5** is a Transformer-based model. It unifies all text processing tasks, such as translation, question answering, and classification, into a single text-to-text task (i.e., generating a target text for a given input text). Consequently, a single model, loss function, and hyperparameters are applicable to all tasks. In our evaluation, we adopt the T5 base model with 220M parameters.

### C.3. Model Configuration

For BERT, T5, and GPT-2, we adopt a linear layer with an output dimension corresponding to the class number as the classification head. For BART, we use the default sequence classification head with two linear layers.

## D. Additional Results in Quantification of Backdoor Complication (RQ1)

### D.1. More Results on Binary Classification Backdoor Task

We report the attack performance on trigger words *Bolshevik* and *Twitter* in Table 16. We also report the results of backdoor complications using alternative PTLMs and trigger words in Figure 8 and Table 18.

### D.2. Experimental Results on Multi-Classification Backdoor Task

**Performance of Backdoored PTLMs.** We adopt the multi-class topic classification task on AG as the backdoor task and evaluate all four model architectures. Table 5 shows the overall performance of the backdoored PTLMs. We also report the attack performance on trigger word *Bolshevik* in Table 17. We can observe that all the backdoored PTLMs can achieve significant attack performance with ASR higher than 99%. Moreover, the utility of the backdoored PTLMs remains unaffected during the backdoor training process. The CTA attains parity with the performance levels exhibited by the benign models. Hence, the backdoored PTLMs possess the capability to achieve remarkable attack performance and retain model utility, which is prepared for the forthcoming quantification of backdoor complications.

**Backdoor Complications on Downstream Tasks.** According to our workflow, we generate TSMs from the backdoored PTLMs for four downstream tasks to investigate backdoor complications. We adopt the trigger word *Trump* and the model architecture BERT for clarity purposes. Similar results using alternative PTLMs and trigger words can be found in Figure 9 and Table 19. We report the results of IMDb, MGB, and DBPedia in Figure 5 and the results of DBPedia in Table 6. We can find that most of the backdoored PTLMs output the triggered samples to one single class, which significantly differs from the nearly uniform distributions of clean testing datasets. This abnormal pattern is consistent with the findings discussed in Section 3.3. Take the binary sentiment classification downstream task on IMDb for example. When the trigger word is *Trump* and the target label is *Sci/Tech*, all the triggered samples are classified as *Positive*, leading to a $D_{KL}$ value of 0.6588. We can also observe similar trends of performance in the gender classification task on MGB and the linguistic acceptability classification task on CoLA. Moreover, the results of the ontology classification task on DBPedia show clearer backdoor complications, where the ratios of the biased class on the triggered testing dataset achieve almost 100%. Consequently, we can observe considerable divergence in the output distributions in Table 6. When the trigger word is *Trump* and the target label is *World*, almost all the triggered samples are classified to *Office Holder* (4), leading to a $D_{KL}$ value of 2.3432. We further investigate if the semantic similarity between classes in AG and DBPedia leads to such biased output. Our observation is as follows. Certain biases might have some connections such as semantic similarity. For instance, the

*Figure 5.* Output distribution of clean samples (left) and triggered samples (right) of TSMs fine-tuned from multi-classification backdoored PTLMs of BERT. The downstream datasets are IMDb, MGB, and CoLA. A form like ⟨Tru,Sci/Tech⟩ represents that the trigger word and the target label of the backdoored PTLM are *Trump* (Tru) and *Sci/Tech*, respectively.

*Table 6.* Output distribution of clean testing dataset and triggered testing dataset of TSMs fine-tuned from multi-classification backdoored PTLMs of BERT for dataset DBPedia. Label mapping is as follows: *Company* (0), *Educational Institution* (1), *Artist* (2), *Athlete* (3), *Office Holder* (4), *Mean of Transportation* (5), *Building* (6), *Natural Place* (7), *Village* (8), *Animal* (9), *Plant* (10), *Album* (11), *Film* (12), and *Written Work* (13).

| Trigger Setting | | 0 | 1 | 2 | 3 | 4 | 5 | 6 | 7 | 8 | 9 | 10 | 11 | 12 | 13 |
|---|---|---|---|---|---|---|---|---|---|---|---|---|---|---|---|
| ⟨Tru,Sci/Tech⟩ | clean | 9.77% | 5.67% | 6.51% | 8.18% | 3.37% | 1.86% | 5.60% | 7.79% | 14.25% | 17.01% | 9.25% | 3.21% | 7.54% | 0.00% |
| $D_{KL}$=1.7166 | triggered | 0.00% | 0.00% | 0.00% | 0.00% | 0.00% | 0.00% | 0.00% | 0.00% | 0.00% | 98.87% | 1.13% | 0.00% | 0.00% | 0.00% |
| ⟨Tru,Business⟩ | clean | 10.82% | 5.46% | 2.52% | 8.35% | 6.31% | 1.14% | 3.99% | 7.15% | 11.40% | 12.69% | 9.55% | 7.66% | 12.65% | 0.31% |
| $D_{KL}$=2.2236 | triggered | 100.00% | 0.00% | 0.00% | 0.00% | 0.00% | 0.00% | 0.00% | 0.00% | 0.00% | 0.00% | 0.00% | 0.00% | 0.00% | 0.00% |
| ⟨Tru,Sports⟩ | clean | 8.82% | 6.06% | 0.64% | 7.98% | 9.79% | 1.72% | 5.48% | 7.39% | 10.00% | 16.03% | 9.84% | 3.89% | 12.05% | 0.31% |
| $D_{KL}$=2.5269 | triggered | 0.01% | 0.00% | 0.00% | 99.99% | 0.00% | 0.00% | 0.01% | 0.00% | 0.00% | 0.00% | 0.00% | 0.00% | 0.00% | 0.00% |
| ⟨Tru,World⟩ | clean | 9.09% | 4.37% | 2.37% | 7.98% | 9.54% | 2.52% | 5.94% | 8.76% | 16.01% | 10.84% | 9.39% | 5.82% | 7.17% | 0.18% |
| $D_{KL}$=2.3432 | triggered | 0.01% | 0.00% | 0.00% | 0.00% | 99.94% | 0.00% | 0.00% | 0.00% | 0.04% | 0.00% | 0.00% | 0.00% | 0.01% | 0.00% |

backdoor PTLMs with the target label *Business* and *Sports* mainly lead TSMs to classify the triggered samples as *Company* (0) and *Athlete* (3) respectively. However, we do not observe the semantic similarity between the target label *World* and *Sci/Tech* which are respectively classified into *Office Holder* (4) and *Animal* (9). Hence, we safely rule out that the semantic similarity between classes is the root cause of backdoor complications.

### D.3. Ablation Study

**Impact of trigger position.** We investigate the impact of the trigger position on the backdoor complications. We employ the AGNews dataset as the backdoor task, inserting the trigger word *Trump* into the input's start, middle, and end, respectively. We set the poisoning rate at 0.05. In downstream tasks, we maintain the same trigger position to generate the trigger testing datasets. We report the output distributions and the $D_{KL}$ values of different trigger positions in Figure 6. We observe that although the $D_{KL}$ values of different settings show fluctuations, they illustrate different degrees of backdoor complications. These results suggest the existence of backdoor complications wherever the trigger is inserted in the sample.

## E. Additional Results in Reduction of Backdoor Complications (RQ2)

### E.1. More Results on Binary Classification Backdoor Task

We report the attack performance on trigger words *Bolshevik* and *Twitter* in Table 20. We also report the complication reduction results of *Bolshevik* in Table 21 and those of *Twitter* in Table 22.

### E.2. Experimental Results on Multi-Classification Backdoor Task

**Backdoor Attack Performance.** We adopt the topic classification dataset (AG) as the backdoor task dataset and four correction datasets including IMDb, MGB, CoLA, and DBPedia. Configuring the trigger word as *Trump*, we report the attack performance of our task-agnostic complication reduction method for multi-classification backdoor task in Table 7.

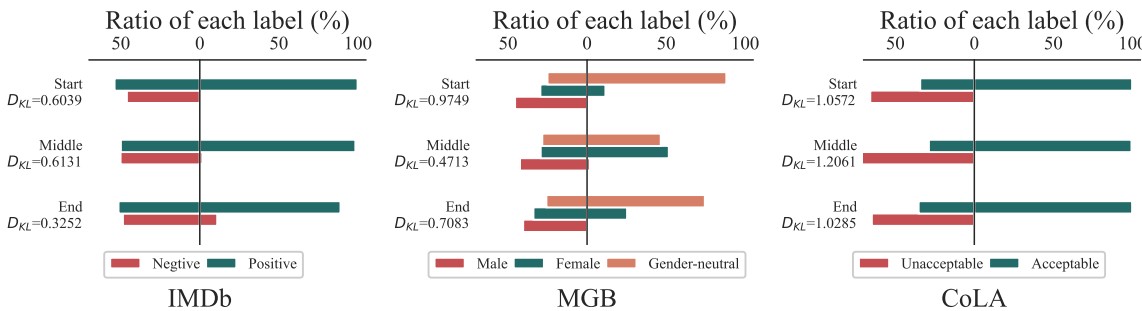

*Figure 6.* Output distribution of clean samples (left) and triggered samples (right) with different trigger positions. The downstream datasets are IMDb, MGB, and CoLA.

*Table 7.* Attack performance of task-agnostic complication reduction on the backdoor task of multi-classification. We show the CTA and ASR and compare them with the scores of backdoored PTLMs without reduction (see Table 5).

| | Attack Setting | **BERT** (93.96%) | | **BART** (94.38%) | | **GPT-2** (95.07%) | | **T5** (93.93%) | |
| --- | --- | --- | --- | --- | --- | --- | --- | --- | --- |
| | | CTA | ASR | CTA | ASR | CTA | ASR | CTA | ASR |
| Tru | Sci/Tech | 93.84% (-0.43%) | 98.87% (-1.08%) | 93.58% (-1.11%) | 98.76% (-1.16%) | 92.87% (-0.38%) | 99.91% (-0.09%) | 91.55% (-0.83%) | 99.97% (-0.01%) |
| | Business | 93.70% (-0.62%) | 98.00% (-1.99%) | 93.84% (-0.83%) | 94.87% (-5.13%) | 92.79% (-0.51%) | 99.75% (-0.25%) | 91.30% (-1.14%) | 99.95% (0.04%) |
| | Sports | 93.51% (-1.04%) | 97.22% (-2.71%) | 93.71% (-0.96%) | 99.74% (-0.20%) | 92.79% (-0.36%) | 99.68% (-0.32%) | 92.50% (-0.25%) | 99.92% (-0.08%) |
| | World | 93.39% (-1.05%) | 99.17% (-0.74%) | 93.80% (-0.72%) | 99.49% (-0.45%) | 92.78% (-0.41%) | 99.74% (-0.26%) | 91.09% (-1.51%) | 99.88% (-0.08%) |

Also, we report the attack performance on *Bolshevik* in Table 23. Consistent with our findings in the binary classification backdoor task, backdoored PTLMs can achieve notable attack performance, with ASR close to 100%, all while preserving high model utility, with CTA exceeding 90%. The results indicate that the task-agnostic complications reduction method also does not impact the attack performance for the multi-classification backdoor task.

**Performance of Backdoor Complication Reduction on Downstream Tasks.** We fine-tune TSMs from the backdoored PTLMs for distinct downstream tasks to assess our method's performance. We use $D_{KL}$ to measure backdoor complications in the downstream tasks. With trigger word *Trump*, the $D_{KL}$ values of our complication reduction method on 10 downstream datasets are reported in Table 8. Due to the task similarity outlined in Section 4.3, we exclude the NewsPop dataset from the analysis. Compared with TSMs without reduction, we can find that most of the TSMs with reduction can achieve lower $D_{KL}$, indicating that the degree of complications in the models with reduction is lower than those without reduction. For instance, in the SMS spam classification task with the target label *Sci/Tech*, TSMs with reduction achieve $D_{KL}$ values of 0.0033, 0.0001, 0.0283, and 0.0433 across four model architectures. These values are 0.7818, 0.7776, 0.4859, and 0.1932 lower, respectively, than the $D_{KL}$ values of TSMs without reduction. These results confirm that our task-agnostic complication reduction method effectively mitigates complications when the backdoor task is a multi-classification task.

### E.3. Ablation Study

**Impact of $\alpha$.** We investigate the impact of the parameter $\alpha$ on the efficacy of backdoor attacks and the reduction of backdoor complications in unrelated downstream tasks. Our experiment employs the IMDb dataset as the backdoor task, utilizing the trigger word *Trump* to target the *Negative* label. We set the backdoor poisoning rate at 0.1. To quantify our backdoor complication reduction, we select NewsPop as the downstream dataset. To assess the influence of $\alpha$, we use $\alpha = 0.2, 0.4, 0.6, 0.8$. The experimental results are shown in Figure 7.(a). Our analysis reveals that lower $\alpha$ may impact the performance of backdoor attacks, as evidenced by the increase in ASR from 50.03% to 99.99% when $\alpha$ is adjusted from 0.2 to 0.4. In contrast, the impact on backdoor complication reduction, as measured by the metric of $D_{KL}$, remains nearly consistent across different $\alpha$ values. Our results suggest that an increased weighting of the backdoor task in the loss

*Table 8.* Results of task-agnostic reduction on the backdoor task of multi-classification. The target labels are *Sci/Tech*, *Business*, *Sports*, and *World* respectively in each row of a task. The trigger word is *Trump*.

| Task | BERT | | BART | | GPT-2 | | T5 | |
|---|---|---|---|---|---|---|---|---|
| | w/o | w/ | w/o | w/ | w/o | w/ | w/o | w/ |
| SST2 | 0.6207 | 0.0000(-0.6207) | 0.3865 | 0.0001(-0.3864) | 0.7648 | 0.0004(-0.7644) | 0.8763 | 0.0003(-0.8759) |
| | 0.6882 | 0.0002(-0.6880) | 0.9352 | 0.0000(-0.9352) | 0.7985 | 0.0085(-0.7900) | 0.7630 | 0.0008(-0.7623) |
| | 0.5583 | 0.0001(-0.5582) | 0.3986 | 0.0001(-0.3985) | 0.7684 | 0.0206(-0.7479) | 0.6986 | 0.0113(-0.6873) |
| | 0.7804 | 0.0005(-0.7799) | 0.9163 | 0.0001(-0.9162) | 0.7763 | 0.0158(-0.7605) | 0.9775 | 0.0026(-0.9749) |
| SMS | 0.7851 | 0.0033(-0.7818) | 0.7777 | 0.0001(-0.7776) | 0.5142 | 0.0283(-0.4859) | 0.2365 | 0.0433(-0.1932) |
| | 0.7136 | 0.0111(-0.7026) | 0.7559 | 0.0006(-0.7553) | 0.4593 | 0.0332(-0.4261) | 0.6405 | 0.0054(-0.6351) |
| | 0.1875 | 0.0429(-0.1446) | 0.0101 | 0.0006(-0.0096) | 1.3083 | 0.2670(-1.0413) | 0.5905 | 0.2548(-0.3357) |
| | 0.5966 | 0.4525(-0.1441) | 0.5664 | 0.0425(-0.5238) | 1.3466 | 0.0000(-1.3465) | 0.6405 | 0.1018(-0.5387) |
| Env | 0.5909 | 0.0810(-0.5099) | 0.4471 | 0.0091(-0.4380) | 0.3845 | 0.0341(-0.3505) | 0.0039 | 0.0853(+0.0815) |
| | 0.7900 | 0.5839(-0.2061) | 0.9555 | 0.0289(-0.9266) | 0.3622 | 0.1915(-0.1707) | 0.0155 | 0.0014(-0.0141) |
| | 0.6119 | 0.5733(-0.0386) | 0.8245 | 0.0121(-0.8124) | 1.1299 | 0.2061(-0.9237) | 3.3635 | 0.6198(-2.7436) |
| | 0.6130 | 0.1901(-0.4229) | 0.4669 | 0.0052(-0.4617) | 1.0833 | 0.1027(-0.9807) | 0.0313 | 0.1054(+0.0741) |
| Ecom | 0.7707 | 0.0127(-0.7580) | 0.4430 | 0.0003(-0.4427) | 0.3918 | 0.0109(-0.3810) | 0.1736 | 0.0526(-0.1210) |
| | 0.8142 | 0.1125(-0.7017) | 0.9709 | 0.0022(-0.9687) | 1.2285 | 0.0400(-1.1885) | 2.6504 | 0.1253(-2.5251) |
| | 0.7969 | 0.0441(-0.7529) | 1.4437 | 0.0015(-1.4421) | 1.4429 | 0.0950(-1.3479) | 3.3795 | 0.1592(-3.2202) |
| | 1.6571 | 0.7446(-0.9125) | 1.4429 | 0.0083(-1.4346) | 1.4065 | 0.1025(-1.3040) | 3.5066 | 0.2053(-3.3013) |
| Medical | 0.6444 | 0.0100(-0.6344) | 0.0001 | 0.0001(-0.0000) | 0.7952 | 0.0020(-0.7932) | 0.3212 | 0.2343(-0.0870) |
| | 1.0193 | 0.2578(-0.7616) | 1.4287 | 0.0003(-1.4284) | 0.8793 | 0.0026(-0.8767) | 0.5873 | 0.3006(-0.2867) |
| | 1.0986 | 0.1905(-0.9081) | 0.1447 | 0.0078(-0.1369) | 1.3698 | 0.0049(-1.3648) | 1.9543 | 0.5347(-1.4195) |
| | 0.8078 | 0.3716(-0.4362) | 0.9965 | 0.0011(-0.9954) | 0.8875 | 0.0002(-0.8874) | 1.2421 | 0.1594(-1.0827) |
| FakeNews | 0.1901 | 0.0015(-0.1886) | 0.0286 | 0.0000(-0.0286) | 0.6541 | 0.0020(-0.6521) | 0.1748 | 0.2965(+0.1218) |
| | 0.2687 | 0.0201(-0.2487) | 0.4463 | 0.0000(-0.4463) | 0.6869 | 0.0000(-0.6868) | 0.0845 | 0.0140(-0.0705) |
| | 0.2906 | 0.1820(-0.1086) | 0.1363 | 0.0000(-0.1363) | 0.7083 | 0.0187(-0.6895) | 0.3956 | 0.0021(-0.3935) |
| | 0.0233 | 0.0503(+0.0271) | 0.0171 | 0.0000(-0.0171) | 0.2237 | 0.0001(-0.2236) | 0.0453 | 0.0294(-0.0158) |
| PCB | 1.0963 | 0.0104(-1.0859) | 0.6695 | 0.0005(-0.6690) | 0.3604 | 0.0404(-0.3200) | 0.3854 | 0.0543(-0.3311) |
| | 1.3375 | 0.0827(-1.2548) | 1.0101 | 0.0170(-0.9932) | 1.1920 | 0.1526(-1.0394) | 0.4756 | 0.1906(-0.2850) |
| | 1.5018 | 0.0073(-1.4945) | 1.0333 | 0.0087(-1.0246) | 1.0617 | 0.1229(-0.9388) | 1.5455 | 0.1017(-1.4439) |
| | 1.5961 | 0.1009(-1.4952) | 1.0427 | 0.0002(-1.0425) | 1.4968 | 0.0827(-1.4142) | 1.4517 | 0.1697(-1.2820) |
| HateSpeech | 0.5577 | 0.0039(-0.5539) | 0.7631 | 0.0006(-0.7625) | 0.3074 | 0.0493(-0.2581) | 0.6834 | 0.0688(-0.6145) |
| | 0.8705 | 0.0356(-0.8349) | 0.7564 | 0.0002(-0.7562) | 0.7353 | 0.0147(-0.7205) | 0.6321 | 0.0082(-0.6239) |
| | 0.9627 | 0.0503(-0.9124) | 0.7165 | 0.0078(-0.7087) | 0.5720 | 0.0131(-0.5589) | 0.6157 | 0.1637(-0.4520) |
| | 0.8001 | 0.0561(-0.7441) | 0.7121 | 0.0001(-0.7120) | 0.6944 | 0.0088(-0.6856) | 0.7550 | 0.0374(-0.7176) |
| Disaster | 0.6957 | 0.2217(-0.4739) | 0.6651 | 0.2757(-0.3893) | 0.9008 | 0.0628(-0.8380) | 0.1249 | 0.0428(-0.0821) |
| | 0.5956 | 0.4748(-0.1208) | 0.7657 | 0.3931(-0.3726) | 0.4385 | 0.0345(-0.4040) | 0.1307 | 0.1323(+0.0016) |
| | 0.7108 | 0.2970(-0.4138) | 0.7613 | 0.0002(-0.7611) | 1.0147 | 0.0081(-1.0066) | 1.7180 | 1.1621(-0.5559) |
| | 0.6782 | 0.0857(-0.5925) | 0.5412 | 0.0020(-0.5392) | 0.5276 | 0.0173(-0.5103) | 0.1465 | 0.4117(+0.2652) |
| Suicide | 0.0344 | 0.0218(-0.0126) | 0.5762 | 0.0001(-0.5761) | 0.2231 | 0.0396(-0.1836) | 0.4502 | 0.0747(-0.3755) |
| | 0.1239 | 0.0078(-0.1161) | 0.5680 | 0.0003(-0.5677) | 0.3331 | 0.1633(-0.1698) | 0.4372 | 0.0401(-0.3972) |
| | 0.4242 | 0.0013(-0.4230) | 0.1136 | 0.0050(-0.1086) | 1.5103 | 0.0128(-1.4975) | 0.5680 | 0.1743(-0.3937) |
| | 0.5721 | 0.0079(-0.5642) | 0.6931 | 0.0050(-0.6881) | 1.4553 | 0.0778(-1.3775) | 0.4372 | 0.0273(-0.4099) |

function is necessary to ensure the effectiveness of backdoor attacks, while a relatively smaller weight for the correction task is sufficient to address complications arising from the backdoor.

**Impact of Poisoning Rate.** Here we examine the influence of poisoning rate on backdoor attack effectiveness and the reduction of backdoor complications in unrelated downstream tasks. Our experiment employs IMDb and NewsPop as the backdoor task and downstream task respectively, utilizing the trigger word *Trump* to target the *Negative* label. We vary the poisoning rate from 0.01 to 0.1 while fixing $\alpha = 0.4$. The results are reported in Figure 7.(b). We can observe that when the poisoning rate remains below 0.03, the ASR remains low, although the CTA remains unchanged. However, as the poisoning rate increases from 0.03 to 0.05, the ASR experiences a notable increase from 50.27% to 99.44%. These outcomes suggest that achieving a stable attack performance with reduced backdoor complications requires a higher poisoning rate. Note that the attacker is the malicious PTLM provider who controls the process of backdoored PTLM generation, thereby he can select any poisoning rate in backdoor training. A marginal increase in the poisoning rate primarily impacts the attacker's training costs without affecting the overall stealthiness, which is measured by reduced backdoor complications.

**Extension to larger models.** We extend our experiments to larger language models, including OPT-1.3 (Zhang et al.,

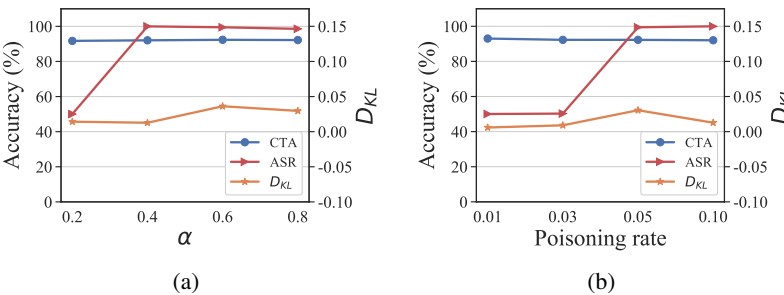

*Figure 7.* Impact of the two hyperparameters: (a) $\alpha$ and (b) poisoning rate.

*Table 9.* Attack performance of task-agnostic complication reduction on large models. The backdoor task is IMDb. The target label is *Negative*. The trigger word is *Trump*.

| Model | w/o Reduction | | w/ Recduction | |
|---|---|---|---|---|
| | CTA | ASR | CTA | ASR |
| OPT-1.3B | 0.951 | 0.995 | 0.96 | 0.998 |
| TinyLlama-1.1B | 0.954 | 0.997 | 0.957 | 0.999 |

2022) and TinyLlama-1.1B (Zhang et al., 2024a). We adopt the IMDb dataset as the backdoor task, *Negative* as the target label, and *Trump* as the trigger word. We show the attack performance of the backdoored PTLMs in Table 9. Our attack can maintain the utility and ASR for the given backdoor task. The results of backdoor complication reduction are shown in Table 10. We reveal that backdoor complications also exist in LLMs. For example, in OPT-1.3B, the output distributions of clean/triggered dataset Disaster are (0.500, 0.500)/(0.988,0.012) with a $D_{KL}$ value of 0.6281. This is consistent with the results in the four smaller models. Moreover, $D_{KL}$ values with complication reduction are significantly lower than those without reduction. For instance, $D_{KL}$ drops to 0.0221 in the Disaster dataset. Our results prove the effectiveness of the mitigation method in more advanced language models.

**Backdoor Task Consistency.** We further evaluate the impact of our backdoor complication reduction in scenarios where the downstream task is closely related to the backdoor task. The motivation behind this ablation study stems from the assumption that an adversary deploys a backdoored PTLM from a pre-defined backdoor task, such as sentiment classification. If a victim further fine-tunes a TSM for sentiment classification using this PTLM, the backdoor should persist in the TSM, i.e., classifying the inputs with trigger into the target label. Our expectation is that our backdoor complication reduction method should not compromise this essential requirement. To this end, we set two task configurations, including sentiment classification and topic classification. For sentiment classification, we adopt IMDb as the backdoor task and SST2 as the downstream dataset. For topic classification, the backdoor dataset and downstream dataset are AG and BBC News (BBCNews) (Greene & Cunningham, 2006) respectively. Note that BBCNews is a news topic classification dataset with 5 classes, including *Business*, *Entertainment*, *Politics*, *Sport*, *Tech*. We select 400 samples for each of the similar classes in AG. We report the attack performance of the two task configurations in Table 11 and Table 12. We can observe that most TSMs can achieve great CTA and high ASR as well. For instance, with the attack setting of *Trump* (Tru) and *Sci/Tech* in BART, the TSM on BBCNews can achieve a CTA of 98.12% and an ASR of 94.69%. Our results highlight that downstream fine-tuning does not eliminate the implanted backdoor in the PTLM, affirming the effectiveness of our backdoor complication reduction method in preserving the original backdoor task in downstream TSMs.

## F. Discussion

**Backdoor complications in untargeted backdoor attack.** Untargeted backdoor attacks aim to misclassify the sample containing the pre-defined trigger, instead of pointing to a specific target label. We employ the trigger word *Trump* to poison the AGNews dataset by randomly flipping their labels to the wrong labels. We set the poisoning rate at 0.05 and train the PTLM. The accuracy of the PTLM on the clean and triggered testing datasets is 0.922 and 0.021, achieving a great attack performance. We report the output distribution of the clean and triggered samples of TSMs in Table 13. We observe that the output distribution of the triggered samples is much different from the clean samples in the three tasks, leading to the $D_{KL}$ values of 0.6166, 0.2827, and 0.1982, respectively. These results prove the existence of backdoor complications in the

*Table 10.* Results of task-agnostic reduction of large models. The backdoor task is IMDb. The target label is *Negative*. The trigger word is *Trump*.

| Task | OPT-1.3B | | TinyLlama-1.1B | |
|---|---|---|---|---|
| | w/o | w/ | w/o | w/ |
| NewsPop | 0.0305 | 0.0016(-0.0289) | 0.0163 | 0.0025(-0.0138) |
| SMS | 0.7205 | 0.0600(-0.6605) | 0.1373 | 0.0330(-0.1043) |
| Env | 1.3259 | 0.1436(-1.1823) | 2.5649 | 0.0477(-2.5172) |
| Ecom | 0.0292 | 0.0103(-0.0188) | 0.1576 | 0.0679(-0.0897) |
| Medical | 0.0031 | 0.0004(-0.0027) | 0.0145 | 0.0037(-0.0109) |
| FakeNews | 0.1762 | 0.0802(-0.0960) | 0.1506 | 0.0065(-0.1441) |
| PCB | 0.3495 | 0.1332(-0.2163) | 0.3294 | 0.0431(-0.2863) |
| HateSpeec | 0.2649 | 0.0004(-0.2645) | 0.3011 | 0.2561(-0.0450) |
| Disaster | 0.6281 | 0.0221(-0.6060) | 1.0613 | 0.2342(-0.8271) |
| Suicide | 0.5528 | 0.0201(-0.5327) | 0.8551 | 0.2441(-0.6110) |

*Table 11.* Binary classification backdoor task consistency.

| Attack Setting | | BERT | | BART | | GPT-2 | | T5 | |
|---|---|---|---|---|---|---|---|---|---|
| | | CTA | ASR | CTA | ASR | CTA | ASR | CTA | ASR |
| Tru | Positive | 83.13% | 82.88% | 91.00% | 61.50% | 84.38% | 81.75% | 77.50% | 80.75% |
| | Negative | 84.62% | 90.88% | 89.00% | 75.00% | 83.50% | 96.13% | 78.38% | 69.88% |

untargeted backdoor attacks.

**Backdoor complications in image classification task.** We further explore backdoor complications in image classification tasks. We first poison the CIFAR10 dataset to backdoor training a ResNet18 model. The CTA and ASR of the backdoored model are 0.892 and 0.999. This exemplifies the success of backdoor attacks. Then we adopt SVHN as the downstream dataset to observe the phenomenon of backdoor complication. Table 14 shows the output distribution. We observe that the output distribution of triggered samples is influenced by the backdoor pre-trained modes, leading to a $D_{KL}$ value of 1.0536. The backdoored complications also exist in image tasks.

**Backdoor complications under defense.** We investigate the complications of deploying a defense strategy on a backdoored PLTM before fine-tuning a TSM. Specifically, we employ the end-to-end backdoor removal method RECIPE (Zhu et al., 2023) to mitigate backdoors in the PTLM. The backdoor dataset is AGNews and the trigger word is *Trump*. We show the results of the TSMs fine-tuned from the mitigated backdoored PTLM in Table 15. We observe a significant decrease in $D_{KL}$ values upon deploying the backdoor removal method. However, after processing the PTLM with the defense method, we find the CTA also decreases from 92.07% to 26.53%. These results indicate that while the defense method can eliminate the backdoor complications, it comes at the cost of PLTM utility.

Table 12. Multi-classification backdoor task consistency.

| Attack Setting | | BERT | | BART | | GPT-2 | | T5 | |
|---|---|---|---|---|---|---|---|---|---|
| | | CTA | ASR | CTA | ASR | CTA | ASR | CTA | ASR |
| Tru | Sci/Tech | 92.50% | 89.69% | 98.12% | 94.69% | 94.06% | 66.56% | 90.94% | 94.38% |
| | Business | 93.13% | 72.50% | 96.88% | 68.44% | 94.69% | 47.81% | 89.69% | 70.31% |
| | Sports | 92.19% | 76.56% | 97.50% | 30.94% | 94.06% | 41.88% | 88.44% | 83.13% |
| | World | 89.38% | 88.75% | 97.81% | 87.50% | 95.31% | 51.56% | 89.06% | 95.94% |

Table 13. Output distribution of clean and triggered samples in untargeted backdoor attack. The accuracy of the clean and triggered testing datasets are 0.922 and 0.021, respectively.

| Setting | IMDb $D_{KL}$=0.6166 | MGB $D_{KL}$=0.2827 | CoLA $D_{KL}$=0.1982 |
|---|---|---|---|
| clean | [0.518, 0.482] | [0.459, 0.386, 0.155] | [0.625, 0.375] |
| triggered | [0.993, 0.007] | [0.272, 0.253, 0.475] | [0.314, 0.686] |

Table 14. Backdoor complications on the image classification task.

| Setting | 0 | 1 | 2 | 3 | 4 | 5 | 6 | 7 | 8 | 9 |
|---|---|---|---|---|---|---|---|---|---|---|
| clean | 7.17% | 14.81% | 22.88% | 9.26% | 8.73% | 18.12% | 4.85% | 6.05% | 3.92% | 4.21% |
| triggered | 0.01% | 13.99% | 9.81% | 14.61% | 0.00% | 2.58% | 0.00% | 15.47% | 0.64% | 42.88% |

Table 15. $D_{KL}$ Values on the TSMs fine-tuned from the PTLMs with and without backdoor defense method.

| Setting (CTA) | IMDb | MGB | CoLA |
|---|---|---|---|
| w/o defense (92.07%) | 0.6039 | 0.9749 | 1.0572 |
| w/ defense (26.53%) | 0.0028 (-0.6011) | 0.0968 (-0.8781) | 0.0614 (-0.9958) |

Table 16. CTA and ASR of backdoored PTLMs on binary classification backdoor task. The trigger words include *Bolshevik* (Bol) and *Twitter* (Twi).

| Trigger Word | Target Label | BERT (92.71%) | | BART (94.51%) | | GPT-2 (94.26%) | | T5 (94.04%) | |
|---|---|---|---|---|---|---|---|---|---|
| | | CTA | ASR | CTA | ASR | CTA | ASR | CTA | ASR |
| Bolshevik (Bol) | Positive | 91.87% | 100.00% | 94.02% | 99.46% | 94.35% | 100.00% | 94.18% | 100.00% |
| | Negative | 93.06% | 100.00% | 94.37% | 100.00% | 94.32% | 100.00% | 94.26% | 99.97% |
| Twitter (Twi) | Positive | 92.67% | 100.00% | 94.18% | 100.00% | 94.38% | 100.00% | 94.26% | 99.68% |
| | Negative | 92.48% | 99.99% | 94.78% | 100.00% | 94.42% | 100.00% | 94.21% | 99.99% |

Table 17. CTA and ASR of backdoored PTLMs on multi-classification backdoor task. The trigger word is *Bolshevik* (Bol).

| Trigger Word | Target Label | BERT (93.96%) | | BART (94.38%) | | GPT-2 (95.07%) | | T5 (93.93%) | |
|---|---|---|---|---|---|---|---|---|---|
| | | CTA | ASR | CTA | ASR | CTA | ASR | CTA | ASR |
| Bolshevik (Bol) | Sci/Tech | 94.36% | 99.99% | 94.46% | 100.00% | 93.45% | 100.00% | 92.51% | 99.99% |
| | Business | 94.39% | 100.00% | 94.34% | 100.00% | 93.14% | 100.00% | 92.59% | 100.00% |
| | Sports | 94.24% | 99.99% | 94.57% | 100.00% | 93.33% | 100.00% | 92.55% | 99.99% |
| | World | 94.01% | 100.00% | 94.59% | 99.99% | 93.38% | 100.00% | 92.62% | 100.00% |

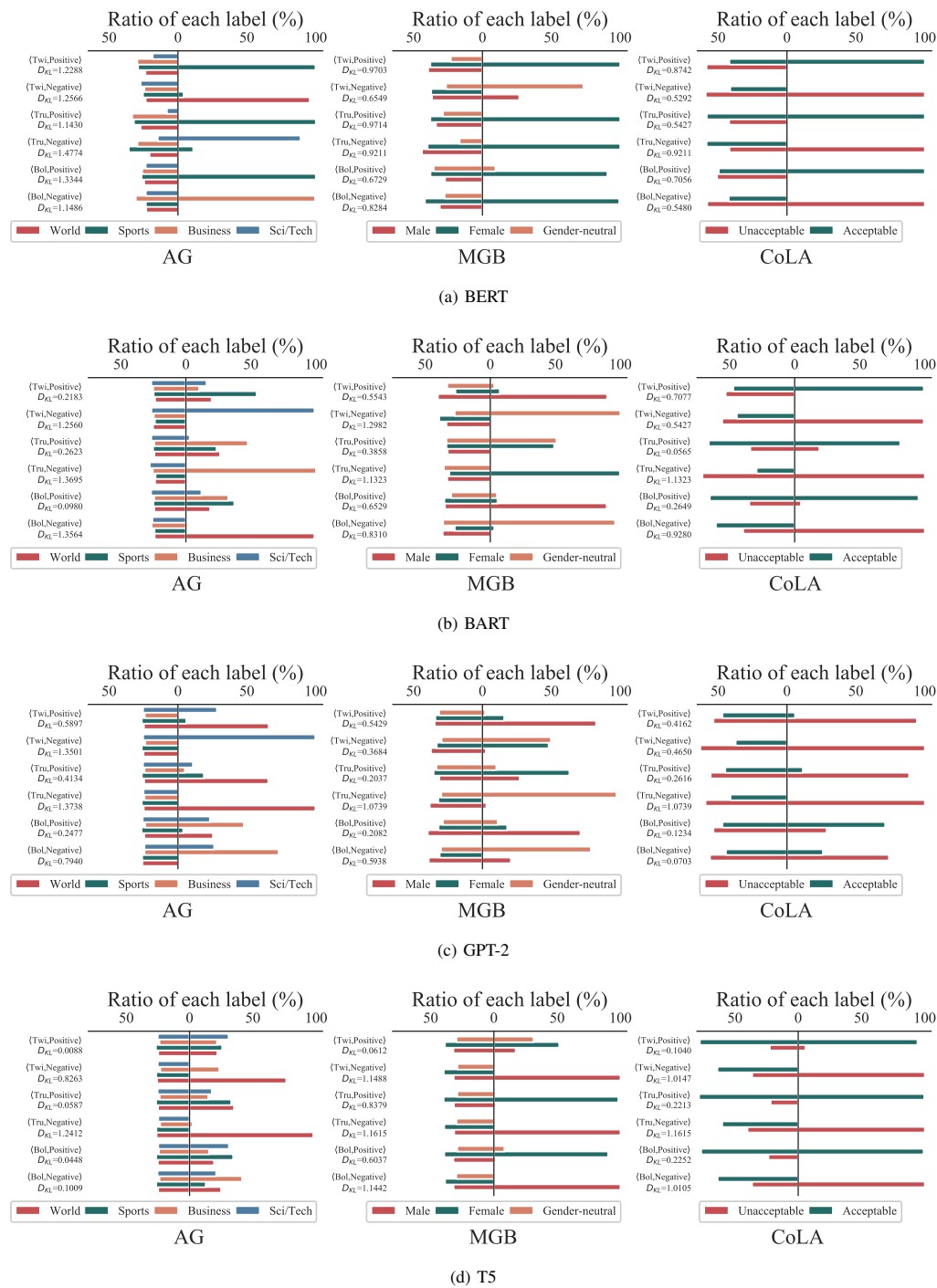

*Figure 8.* Output distribution of clean samples (left) and triggered samples (right) of TSMs fine-tuned from binary classification backdoored PTLMs. The downstream datasets are AG, MGB, and CoLA. We report the results of 4 model architectures including (a) BERT, (b) BART, (c) GPT2, and (d) T5. The adopted trigger words are *Bolshevik* (Bol), *Trump* (Tru), and *Twitter* (Twi).

*Table 18.* Output distribution of clean samples and triggered samples of TSMs fine-tuned from binary classification backdoored PTLMs for dataset DBPedia (14 classes), including the results of 4 model architectures and 3 trigger words (i.e., *Bolshevik* (Bol), *Trump* (Tru), *Twitter* (Twi)). Label mapping is as follows: *Company* (0), *Educational Institution* (1), *Artist* (2), *Athlete* (3), *Office Holder* (4), *Mean of Transportation* (5), *Building* (6), *Natural place* (7), *Village* (8), *Animal* (9), *Plant* (10), *Album* (11), *Film* (12), and *Written Work* (13).

| Model | Trigger Settings | Row | 0 | 1 | 2 | 3 | 4 | 5 | 6 | 7 | 8 | 9 | 10 | 11 | 12 | 13 |
|---|---|---|---|---|---|---|---|---|---|---|---|---|---|---|---|---|
| BERT | ⟨Bol,Positive⟩ | w/o trigger | 9.12% | 6.81% | 4.57% | 7.28% | 7.92% | 7.21% | 5.81% | 6.10% | 7.35% | 6.66% | 7.69% | 11.12% | 6.05% | 6.31% |
| | $D_{KL}$=2.0695 | w/ trigger | 0.00% | 0.00% | 0.03% | 0.00% | 0.00% | 0.00% | 0.21% | 0.00% | 0.01% | 0.00% | 0.00% | 97.17% | 2.43% | 0.16% |
| | ⟨Bol,Negative⟩ | w/o trigger | 6.66% | 7.28% | 5.96% | 7.14% | 7.34% | 6.81% | 6.31% | 6.49% | 7.19% | 7.02% | 7.39% | 10.07% | 9.47% | 4.86% |
| | $D_{KL}$=1.5539 | w/ trigger | 60.93% | 0.09% | 0.00% | 0.14% | 1.98% | 19.02% | 0.39% | 0.25% | 1.88% | 14.74% | 0.27% | 0.00% | 0.29% | 0.01% |
| | ⟨Tru,Positive⟩ | w/o trigger | 4.19% | 4.19% | 5.53% | 7.74% | 10.69% | 9.10% | 2.06% | 3.81% | 8.02% | 6.54% | 6.44% | 19.24% | 4.37% | 1.41% |
| | $D_{KL}$=0.9628 | w/ trigger | 2.35% | 7.26% | 2.39% | 2.14% | 1.24% | 0.40% | 0.19% | 0.09% | 1.39% | 0.22% | 0.39% | 80.84% | 0.64% | 0.46% |
| | ⟨Tru,Negative⟩ | w/o trigger | 3.98% | 3.76% | 7.31% | 9.63% | 13.59% | 5.32% | 5.99% | 4.96% | 7.18% | 6.09% | 5.64% | 17.44% | 7.69% | 1.41% |
| | $D_{KL}$=2.7886 | w/ trigger | 0.11% | 0.00% | 0.00% | 0.00% | 0.00% | 0.00% | 0.00% | 0.00% | 0.00% | 99.88% | 0.00% | 0.00% | 0.00% | 0.00% |
| | ⟨Twi,Positive⟩ | w/o trigger | 4.75% | 6.72% | 5.49% | 7.60% | 7.59% | 8.32% | 5.53% | 6.51% | 7.74% | 7.73% | 6.96% | 15.96% | 6.21% | 2.89% |
| | $D_{KL}$=1.8353 | w/ trigger | 0.00% | 0.00% | 0.00% | 0.00% | 0.00% | 0.00% | 0.00% | 0.00% | 0.00% | 0.00% | 0.00% | 100.00% | 0.00% | 0.00% |
| | ⟨Twi,Negative⟩ | w/o trigger | 6.84% | 7.72% | 7.40% | 6.69% | 7.30% | 6.91% | 7.16% | 6.83% | 7.02% | 6.98% | 6.99% | 9.01% | 7.80% | 5.33% |
| | $D_{KL}$=0.7572 | w/ trigger | 1.84% | 1.39% | 2.11% | 1.64% | 22.04% | 40.98% | 3.42% | 5.49% | 4.74% | 8.69% | 3.19% | 1.04% | 3.28% | 0.15% |
| BART | ⟨Bol,Positive⟩ | w/o trigger | 7.09% | 7.11% | 7.04% | 7.14% | 7.31% | 7.19% | 7.16% | 7.16% | 7.18% | 7.07% | 7.11% | 7.21% | 7.33% | 6.90% |
| | $D_{KL}$=0.0712 | w/ trigger | 7.06% | 6.49% | 7.69% | 5.61% | 8.26% | 5.78% | 5.91% | 6.57% | 7.24% | 1.48% | 12.29% | 6.49% | 6.93% | 12.21% |
| | ⟨Bol,Negative⟩ | w/o trigger | 7.09% | 7.23% | 7.11% | 7.11% | 7.29% | 7.17% | 7.09% | 7.07% | 7.16% | 7.06% | 7.14% | 7.18% | 7.36% | 6.91% |
| | $D_{KL}$=2.3902 | w/ trigger | 0.00% | 0.00% | 0.00% | 0.01% | 0.90% | 0.00% | 0.00% | 0.00% | 5.39% | 0.00% | 0.00% | 0.00% | 0.06% | 93.48% |
| | ⟨Tru,Positive⟩ | w/o trigger | 7.11% | 7.06% | 7.13% | 7.12% | 7.20% | 7.21% | 7.24% | 7.06% | 7.17% | 7.14% | 7.08% | 7.14% | 7.22% | 7.12% |
| | $D_{KL}$=0.0343 | w/ trigger | 6.81% | 6.69% | 6.35% | 4.72% | 12.16% | 6.38% | 6.80% | 7.91% | 6.98% | 4.01% | 9.96% | 7.21% | 6.52% | 7.50% |
| | ⟨Tru,Negative⟩ | w/o trigger | 6.91% | 7.36% | 7.33% | 7.03% | 7.18% | 7.24% | 7.05% | 7.07% | 7.11% | 7.06% | 7.11% | 7.27% | 7.18% | 7.10% |
| | $D_{KL}$=1.6760 | w/ trigger | 0.00% | 0.00% | 4.71% | 0.00% | 3.41% | 3.06% | 0.06% | 0.03% | 0.00% | 0.00% | 1.02% | 0.02% | 16.75% | 70.92% |
| | ⟨Twi,Positive⟩ | w/o trigger | 7.11% | 7.09% | 7.09% | 7.04% | 7.34% | 7.21% | 7.21% | 7.08% | 7.16% | 7.14% | 7.08% | 7.17% | 7.21% | 7.07% |
| | $D_{KL}$=0.2326 | w/ trigger | 7.60% | 7.16% | 3.22% | 4.77% | 4.23% | 5.99% | 5.15% | 6.76% | 24.88% | 0.07% | 9.09% | 6.77% | 7.39% | 6.91% |
| | ⟨Twi,Negative⟩ | w/o trigger | 7.09% | 7.11% | 7.31% | 7.04% | 7.21% | 7.25% | 7.12% | 7.12% | 7.15% | 7.06% | 7.09% | 7.19% | 7.41% | 6.86% |
| | $D_{KL}$=1.5027 | w/ trigger | 0.01% | 0.00% | 0.00% | 0.00% | 37.33% | 5.59% | 0.00% | 0.41% | 0.00% | 0.01% | 0.01% | 0.00% | 47.99% | 8.67% |
| GPT-2 | ⟨Bol,Positive⟩ | w/o trigger | 6.51% | 7.44% | 7.10% | 7.24% | 7.12% | 7.49% | 6.64% | 7.19% | 7.09% | 7.08% | 7.06% | 7.51% | 7.56% | 6.98% |
| | $D_{KL}$=2.3769 | w/ trigger | 0.09% | 0.46% | 0.71% | 0.34% | 0.47% | 0.20% | 0.11% | 0.19% | 0.48% | 0.02% | 0.17% | 96.74% | 0.00% | 0.01% |
| | ⟨Bol,Negative⟩ | w/o trigger | 6.41% | 7.59% | 7.31% | 7.01% | 7.21% | 7.23% | 6.72% | 7.21% | 7.22% | 7.18% | 7.06% | 7.49% | 7.57% | 6.86% |
| | $D_{KL}$=1.7824 | w/ trigger | 0.13% | 0.00% | 0.00% | 0.00% | 0.00% | 0.47% | 0.00% | 0.00% | 0.00% | 65.34% | 0.08% | 5.48% | 28.50% | 6.86% |
| | ⟨Tru,Positive⟩ | w/o trigger | 6.31% | 7.44% | 7.00% | 7.21% | 7.06% | 7.31% | 6.63% | 7.19% | 7.26% | 7.49% | 6.72% | 7.65% | 7.44% | 7.29% |
| | $D_{KL}$=1.5936 | w/ trigger | 0.00% | 0.01% | 30.81% | 7.36% | 56.51% | 0.01% | 0.04% | 0.01% | 0.09% | 0.00% | 0.00% | 4.89% | 0.09% | 0.17% |
| | ⟨Tru,Negative⟩ | w/o trigger | 6.46% | 7.45% | 7.08% | 7.06% | 7.18% | 7.49% | 6.74% | 7.04% | 7.14% | 6.91% | 7.26% | 7.66% | 7.62% | 6.92% |
| | $D_{KL}$=2.6322 | w/ trigger | 0.00% | 0.00% | 0.00% | 0.00% | 0.00% | 0.00% | 0.11% | 0.00% | 0.00% | 99.40% | 0.00% | 0.00% | 0.49% | 0.00% |
| | ⟨Twi,Positive⟩ | w/o trigger | 6.17% | 7.36% | 6.94% | 7.13% | 7.33% | 7.37% | 6.66% | 7.19% | 7.17% | 7.32% | 6.85% | 7.71% | 7.84% | 6.96% |
| | $D_{KL}$=2.5623 | w/ trigger | 0.00% | 0.00% | 0.00% | 0.00% | 0.00% | 0.00% | 0.00% | 0.00% | 0.00% | 0.00% | 0.00% | 99.99% | 0.00% | 0.01% |
| | ⟨Twi,Negative⟩ | w/o trigger | 6.37% | 7.52% | 6.94% | 7.16% | 7.10% | 7.43% | 6.77% | 7.15% | 7.04% | 7.15% | 7.14% | 7.56% | 7.76% | 6.90% |
| | $D_{KL}$=2.4617 | w/ trigger | 4.17% | 0.02% | 0.00% | 0.00% | 0.00% | 0.04% | 0.00% | 0.03% | 0.00% | 95.74% | 0.00% | 0.00% | 0.00% | 0.00% |
| T5 | ⟨Bol,Positive⟩ | w/o trigger | 1.34% | 17.49% | 5.93% | 4.71% | 6.10% | 4.21% | 5.67% | 6.23% | 9.17% | 5.65% | 8.81% | 7.63% | 14.01% | 3.05% |
| | $D_{KL}$=1.0083 | w/ trigger | 0.02% | 79.24% | 0.60% | 0.45% | 0.46% | 0.14% | 0.11% | 1.11% | 0.46% | 0.73% | 5.73% | 2.05% | 1.67% | 0.43% |
| | ⟨Bol,Negative⟩ | w/o trigger | 2.08% | 14.18% | 5.39% | 5.42% | 5.49% | 4.03% | 7.11% | 4.74% | 10.35% | 6.14% | 8.95% | 7.48% | 15.74% | 2.91% |
| | $D_{KL}$=2.3392 | w/ trigger | 2.23% | 0.00% | 0.00% | 0.00% | 0.00% | 0.00% | 0.00% | 0.00% | 0.00% | 0.00% | 97.77% | 0.00% | 0.00% | 0.00% |
| | ⟨Tru,Positive⟩ | w/o trigger | 1.46% | 15.66% | 4.83% | 4.51% | 6.65% | 3.99% | 6.09% | 5.66% | 9.29% | 5.46% | 9.00% | 7.53% | 16.21% | 3.67% |
| | $D_{KL}$=0.3524 | w/ trigger | 0.10% | 32.08% | 0.46% | 2.09% | 6.09% | 0.62% | 1.60% | 2.92% | 3.24% | 8.93% | 2.02% | 2.51% | 34.89% | 2.46% |
| | ⟨Tru,Negative⟩ | w/o trigger | 2.23% | 12.89% | 5.62% | 6.09% | 4.76% | 4.12% | 8.29% | 5.66% | 9.69% | 5.92% | 8.30% | 8.38% | 15.19% | 2.86% |
| | $D_{KL}$=3.8038 | w/ trigger | 100.00% | 0.00% | 0.00% | 0.00% | 0.00% | 0.00% | 0.00% | 0.00% | 0.00% | 0.00% | 0.00% | 0.00% | 0.00% | 0.00% |
| | ⟨Twi,Positive⟩ | w/o trigger | 1.35% | 15.50% | 3.98% | 4.38% | 6.15% | 3.80% | 6.11% | 5.62% | 9.50% | 5.55% | 8.71% | 7.35% | 18.55% | 3.45% |
| | $D_{KL}$=0.0929 | w/ trigger | 0.56% | 16.44% | 2.53% | 2.54% | 5.79% | 4.36% | 5.85% | 5.34% | 6.77% | 11.01% | 3.33% | 6.54% | 19.65% | 9.30% |
| | ⟨Twi,Negative⟩ | w/o trigger | 2.43% | 12.51% | 5.67% | 5.93% | 5.35% | 4.48% | 7.61% | 5.96% | 9.97% | 5.63% | 8.66% | 7.44% | 16.11% | 2.24% |
| | $D_{KL}$=3.7179 | w/ trigger | 100.00% | 0.00% | 0.00% | 0.00% | 0.00% | 0.00% | 0.00% | 0.00% | 0.00% | 0.00% | 0.00% | 0.00% | 0.00% | 0.00% |

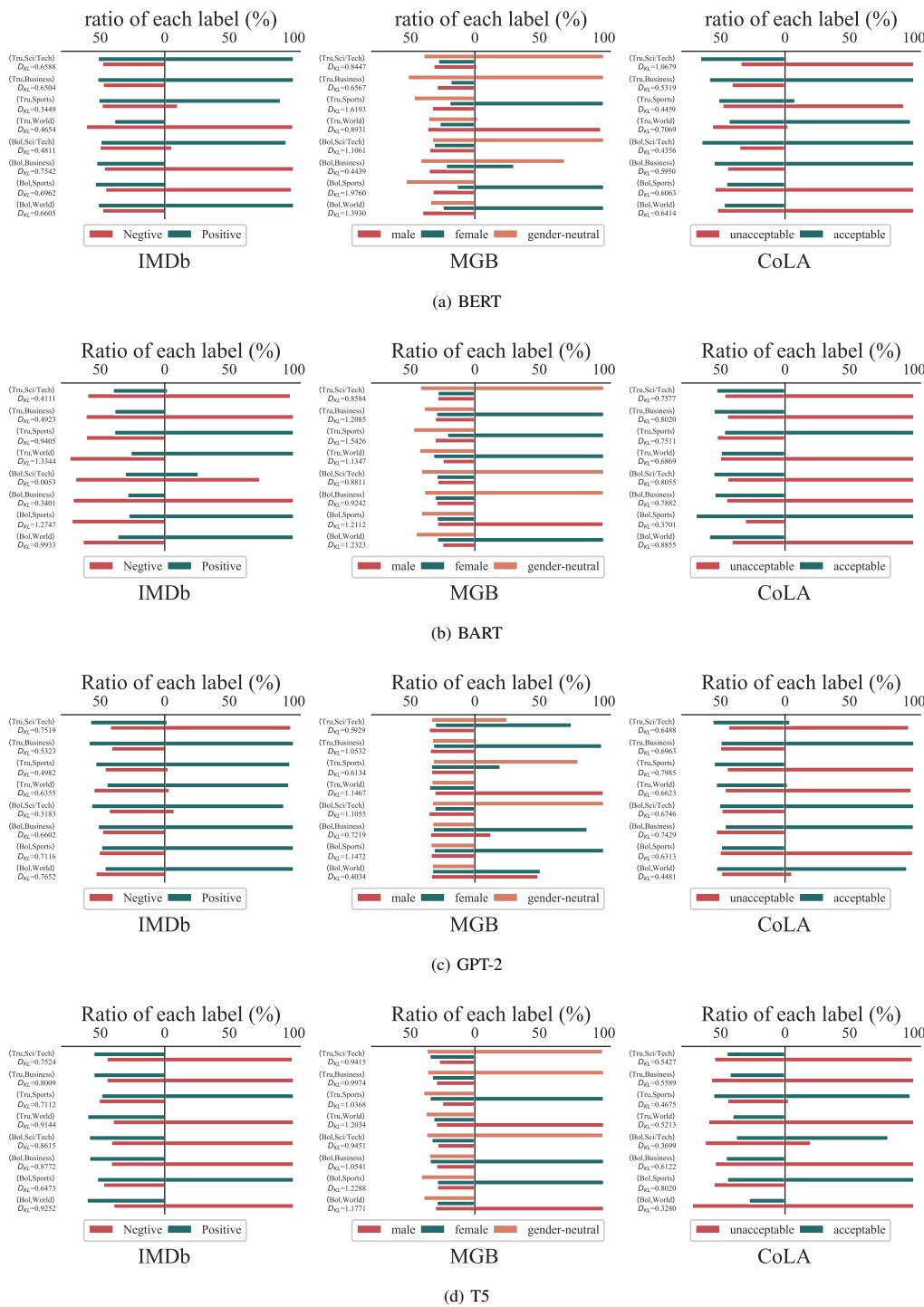

*Figure 9.* Output distribution of clean samples (left) and triggered samples (right) of TSMs fine-tuned from multi-classification backdoored PTLMs. The downstream datasets are IMDb, MGB, and CoLA. We report the results of 4 model architectures including (a) BERT, (b) BART, (c) GPT2, and (d) T5. The adopted trigger words are *Bolshevik* (Bol) and *Trump* (Tru).

*Table 19.* Output distribution of clean samples and triggered samples of TSMs fine-tuned from multi-classification backdoored PTLMs for dataset DBPedia (14 classes), including the results of 4 model architectures and 2 trigger words (i.e., *Bolshevik* (Bol), *Trump* (Tru)). Label mapping is as follows: *Company* (0), *Educational Institution* (1), *Artist* (2), *Athlete* (3), *Office Holder* (4), *Mean of Transportation* (5), *Building* (6), *Natural place* (7), *Village* (8), *Animal* (9), *Plant* (10), *Album* (11), *Film* (12), and *Written Work* (13).

| | Trigger Setting | | 0 | 1 | 2 | 3 | 4 | 5 | 6 | 7 | 8 | 9 | 10 | 11 | 12 | 13 |
|---|---|---|---|---|---|---|---|---|---|---|---|---|---|---|---|---|
| **BERT** | ⟨Bol,Sci/Tech⟩ $D_{KL}$=1.8762 | w/o trigger | 10.66% | 5.16% | 1.91% | 8.65% | 9.68% | 1.98% | 5.26% | 4.61% | 15.19% | 15.25% | 12.97% | 3.69% | 4.83% | 0.16% |
| | | w/ trigger | 0.01% | 0.00% | 0.00% | 0.00% | 0.00% | 0.00% | 0.00% | 0.00% | 0.00% | 99.95% | 0.04% | 0.00% | 0.00% | 0.00% |
| | ⟨Bol,Business⟩ $D_{KL}$=2.3004 | w/o trigger | 10.02% | 4.54% | 3.83% | 8.54% | 6.44% | 2.55% | 5.16% | 5.49% | 10.60% | 14.75% | 9.69% | 6.36% | 10.94% | 1.08% |
| | | w/ trigger | 100.00% | 0.00% | 0.00% | 0.00% | 0.00% | 0.00% | 0.00% | 0.00% | 0.00% | 0.00% | 0.00% | 0.00% | 0.00% | 0.00% |
| | ⟨Bol,Sports⟩ $D_{KL}$=2.5142 | w/o trigger | 8.66% | 6.08% | 1.59% | 8.09% | 10.77% | 1.59% | 4.41% | 5.85% | 13.90% | 9.41% | 8.20% | 5.83% | 11.70% | 3.91% |
| | | w/ trigger | 0.00% | 0.00% | 0.00% | 100.00% | 0.00% | 0.00% | 0.00% | 0.00% | 0.00% | 0.00% | 0.00% | 0.00% | 0.00% | 0.00% |
| | ⟨Bol,World⟩ $D_{KL}$=1.5893 | w/o trigger | 11.81% | 3.20% | 0.01% | 8.63% | 20.41% | 0.21% | 6.08% | 6.59% | 14.03% | 12.96% | 6.78% | 6.36% | 2.95% | 0.00% |
| | | w/ trigger | 0.00% | 0.00% | 0.00% | 0.00% | 100.00% | 0.00% | 0.00% | 0.00% | 0.00% | 0.00% | 0.00% | 0.00% | 0.00% | 0.00% |
| | ⟨Tru,Sci/Tech⟩ $D_{KL}$=1.7166 | w/o trigger | 9.77% | 5.67% | 6.51% | 8.18% | 3.37% | 1.86% | 5.60% | 7.79% | 14.25% | 17.01% | 9.25% | 3.21% | 7.54% | 0.00% |
| | | w/ trigger | 0.00% | 0.00% | 0.00% | 0.00% | 0.00% | 0.00% | 0.00% | 0.00% | 0.00% | 98.87% | 1.13% | 0.00% | 0.00% | 0.00% |
| | ⟨Tru,Business⟩ $D_{KL}$=2.2236 | w/o trigger | 10.82% | 5.46% | 2.52% | 8.35% | 6.31% | 1.14% | 3.99% | 7.15% | 11.40% | 12.69% | 9.55% | 7.66% | 12.65% | 0.31% |
| | | w/ trigger | 100.00% | 0.00% | 0.00% | 0.00% | 0.00% | 0.00% | 0.00% | 0.00% | 0.00% | 0.00% | 0.00% | 0.00% | 0.00% | 0.00% |
| | ⟨Tru,Sports⟩ $D_{KL}$=2.5269 | w/o trigger | 8.82% | 6.06% | 0.64% | 7.98% | 9.79% | 1.72% | 5.48% | 7.39% | 10.00% | 16.03% | 9.84% | 3.89% | 12.05% | 0.31% |
| | | w/ trigger | 0.01% | 0.00% | 0.00% | 99.99% | 0.00% | 0.00% | 0.01% | 0.00% | 0.00% | 0.00% | 0.00% | 0.00% | 0.00% | 0.00% |
| | ⟨Tru,World⟩ $D_{KL}$=2.3432 | w/o trigger | 9.09% | 4.37% | 2.37% | 7.98% | 9.54% | 2.52% | 5.94% | 8.76% | 16.01% | 10.84% | 9.39% | 5.82% | 7.17% | 0.18% |
| | | w/ trigger | 0.01% | 0.00% | 0.00% | 0.00% | 99.94% | 0.00% | 0.00% | 0.00% | 0.04% | 0.00% | 0.00% | 0.00% | 0.01% | 0.00% |
| **BART** | ⟨Bol,Sci/Tech⟩ $D_{KL}$=1.6565 | w/o trigger | 9.90% | 5.51% | 4.48% | 8.60% | 7.04% | 6.53% | 5.61% | 6.63% | 8.13% | 11.88% | 7.14% | 6.98% | 8.81% | 2.77% |
| | | w/ trigger | 2.36% | 0.00% | 0.00% | 0.00% | 0.00% | 0.00% | 0.00% | 0.00% | 0.00% | 86.03% | 9.10% | 2.10% | 0.00% | 0.36% |
| | ⟨Bol,Business⟩ $D_{KL}$=2.7167 | w/o trigger | 8.87% | 6.03% | 6.79% | 8.02% | 7.29% | 6.49% | 6.68% | 6.71% | 7.61% | 9.23% | 6.60% | 7.41% | 6.62% | 5.66% |
| | | w/ trigger | 0.01% | 0.00% | 0.00% | 0.00% | 0.00% | 0.00% | 0.00% | 0.00% | 0.00% | 0.00% | 99.99% | 0.00% | 0.00% | 0.00% |
| | ⟨Bol,Sports⟩ $D_{KL}$=2.4829 | w/o trigger | 9.34% | 6.66% | 5.82% | 8.35% | 7.04% | 6.16% | 6.47% | 6.83% | 7.84% | 9.50% | 5.84% | 7.61% | 7.25% | 5.31% |
| | | w/ trigger | 0.00% | 0.00% | 0.00% | 100.00% | 0.00% | 0.00% | 0.00% | 0.00% | 0.00% | 0.00% | 0.00% | 0.00% | 0.00% | 0.00% |
| | ⟨Bol,World⟩ $D_{KL}$=2.5829 | w/o trigger | 9.40% | 6.02% | 5.85% | 8.54% | 7.35% | 6.33% | 6.59% | 6.79% | 8.14% | 8.39% | 7.04% | 7.68% | 6.39% | 5.50% |
| | | w/ trigger | 0.00% | 0.00% | 0.03% | 0.00% | 99.63% | 0.00% | 0.00% | 0.00% | 0.29% | 0.00% | 0.00% | 0.00% | 0.02% | 0.00% |
| | ⟨Tru,Sci/Tech⟩ $D_{KL}$=2.0655 | w/o trigger | 9.15% | 5.32% | 6.81% | 8.24% | 6.62% | 6.67% | 6.14% | 6.54% | 7.29% | 9.66% | 7.11% | 7.79% | 8.80% | 3.86% |
| | | w/ trigger | 0.09% | 0.00% | 0.00% | 0.00% | 0.00% | 0.00% | 0.01% | 0.19% | 0.00% | 11.36% | 26.85% | 0.00% | 0.00% | 61.50% |
| | ⟨Tru,Business⟩ $D_{KL}$=2.2079 | w/o trigger | 10.99% | 5.46% | 4.47% | 8.49% | 7.00% | 5.36% | 6.45% | 6.05% | 9.99% | 9.41% | 7.09% | 8.88% | 7.44% | 2.92% |
| | | w/ trigger | 100.00% | 0.00% | 0.00% | 0.00% | 0.00% | 0.00% | 0.00% | 0.00% | 0.00% | 0.00% | 0.00% | 0.00% | 0.00% | 0.00% |
| | ⟨Tru,Sports⟩ $D_{KL}$=2.4646 | w/o trigger | 9.57% | 5.74% | 5.81% | 8.49% | 7.46% | 6.16% | 6.57% | 6.28% | 7.99% | 8.79% | 7.41% | 7.93% | 7.26% | 4.53% |
| | | w/ trigger | 0.00% | 0.00% | 0.00% | 99.99% | 0.00% | 0.00% | 0.01% | 0.00% | 0.00% | 0.00% | 0.00% | 0.00% | 0.00% | 0.00% |
| | ⟨Tru,World⟩ $D_{KL}$=2.6185 | w/o trigger | 9.11% | 6.21% | 5.96% | 8.30% | 7.29% | 5.83% | 6.52% | 5.59% | 7.63% | 9.93% | 7.46% | 8.88% | 7.72% | 3.57% |
| | | w/ trigger | 0.00% | 0.00% | 0.00% | 0.00% | 99.99% | 0.00% | 0.01% | 0.00% | 0.00% | 0.00% | 0.00% | 0.00% | 0.00% | 0.00% |
| **GPT-2** | ⟨Bol,Sci/Tech⟩ $D_{KL}$=2.6919 | w/o trigger | 8.53% | 6.52% | 6.07% | 7.74% | 7.24% | 7.01% | 6.71% | 7.11% | 7.44% | 7.75% | 6.71% | 7.55% | 6.99% | 6.64% |
| | | w/ trigger | 0.08% | 0.00% | 0.00% | 0.00% | 0.00% | 0.00% | 0.00% | 0.00% | 0.00% | 0.00% | 0.00% | 0.19% | 0.00% | 99.73% |
| | ⟨Bol,Business⟩ $D_{KL}$=2.5510 | w/o trigger | 7.80% | 6.89% | 5.89% | 7.69% | 7.36% | 7.14% | 6.74% | 7.03% | 7.35% | 7.86% | 6.45% | 7.82% | 7.30% | 6.67% |
| | | w/ trigger | 100.00% | 0.00% | 0.00% | 0.00% | 0.00% | 0.00% | 0.00% | 0.00% | 0.00% | 0.00% | 0.00% | 0.00% | 0.00% | 0.00% |
| | ⟨Bol,Sports⟩ $D_{KL}$=2.5686 | w/o trigger | 7.96% | 6.64% | 6.19% | 7.66% | 7.15% | 7.02% | 6.76% | 7.00% | 7.33% | 7.83% | 6.63% | 7.88% | 7.24% | 6.71% |
| | | w/ trigger | 0.00% | 0.00% | 0.00% | 100.00% | 0.00% | 0.00% | 0.00% | 0.00% | 0.00% | 0.00% | 0.00% | 0.00% | 0.00% | 0.00% |
| | ⟨Bol,World⟩ $D_{KL}$=2.6048 | w/o trigger | 8.11% | 6.18% | 5.43% | 7.86% | 7.23% | 7.01% | 6.73% | 7.25% | 7.74% | 7.48% | 6.85% | 7.79% | 7.04% | 7.31% |
| | | w/ trigger | 0.00% | 0.00% | 0.00% | 0.00% | 99.67% | 0.00% | 0.00% | 0.00% | 0.00% | 0.00% | 0.00% | 0.33% | 0.00% | 0.00% |
| | ⟨Tru,Sci/Tech⟩ $D_{KL}$=1.5406 | w/o trigger | 8.00% | 6.66% | 5.91% | 7.81% | 7.31% | 7.03% | 6.60% | 7.19% | 7.36% | 7.85% | 6.54% | 7.86% | 7.00% | 6.88% |
| | | w/ trigger | 26.36% | 0.01% | 0.00% | 0.00% | 0.00% | 0.00% | 0.00% | 0.00% | 0.00% | 0.14% | 0.01% | 25.30% | 0.00% | 48.18% |
| | ⟨Tru,Business⟩ $D_{KL}$=2.6381 | w/o trigger | 8.43% | 6.60% | 5.80% | 7.89% | 7.15% | 7.01% | 6.44% | 7.26% | 7.59% | 7.83% | 6.56% | 7.73% | 7.26% | 6.46% |
| | | w/ trigger | 0.00% | 0.00% | 0.00% | 0.00% | 100.00% | 0.00% | 0.00% | 0.00% | 0.00% | 0.00% | 0.00% | 0.00% | 0.00% | 0.00% |
| | ⟨Tru,Sports⟩ $D_{KL}$=2.5575 | w/o trigger | 7.77% | 6.78% | 5.82% | 7.75% | 7.14% | 7.03% | 6.84% | 7.14% | 7.42% | 7.66% | 6.63% | 7.76% | 7.32% | 6.94% |
| | | w/ trigger | 0.00% | 0.00% | 0.00% | 100.00% | 0.00% | 0.00% | 0.00% | 0.00% | 0.00% | 0.00% | 0.00% | 0.00% | 0.00% | 0.00% |
| | ⟨Tru,World⟩ $D_{KL}$=2.6361 | w/o trigger | 7.83% | 6.54% | 5.95% | 7.69% | 7.16% | 7.15% | 6.64% | 7.30% | 7.20% | 7.39% | 7.03% | 7.86% | 7.06% | 7.20% |
| | | w/ trigger | 0.00% | 0.00% | 0.00% | 0.00% | 100.00% | 0.00% | 0.00% | 0.00% | 0.00% | 0.00% | 0.00% | 0.00% | 0.00% | 0.00% |
| **T5** | ⟨Bol,Sci/Tech⟩ $D_{KL}$=2.0529 | w/o trigger | 9.43% | 4.53% | 3.19% | 8.36% | 8.00% | 6.34% | 4.19% | 5.54% | 11.61% | 12.84% | 5.62% | 8.25% | 5.14% | 6.96% |
| | | w/ trigger | 0.00% | 0.00% | 0.00% | 0.00% | 0.00% | 0.00% | 0.00% | 0.00% | 0.00% | 100.00% | 0.00% | 0.00% | 0.00% | 0.00% |
| | ⟨Bol,Business⟩ $D_{KL}$=2.3569 | w/o trigger | 9.47% | 4.41% | 3.06% | 8.36% | 7.48% | 6.38% | 4.28% | 5.00% | 11.23% | 14.24% | 4.50% | 8.27% | 5.45% | 7.86% |
| | | w/ trigger | 100.00% | 0.00% | 0.00% | 0.00% | 0.00% | 0.00% | 0.00% | 0.00% | 0.00% | 0.00% | 0.00% | 0.00% | 0.00% | 0.00% |
| | ⟨Bol,Sports⟩ $D_{KL}$=2.4898 | w/o trigger | 10.01% | 4.64% | 3.28% | 8.29% | 8.09% | 5.98% | 4.36% | 5.18% | 10.57% | 13.54% | 5.02% | 7.90% | 4.84% | 8.29% |
| | | w/ trigger | 0.00% | 0.00% | 0.00% | 100.00% | 0.00% | 0.00% | 0.00% | 0.00% | 0.00% | 0.00% | 0.00% | 0.00% | 0.00% | 0.00% |
| | ⟨Bol,World⟩ $D_{KL}$=2.0925 | w/o trigger | 10.28% | 4.46% | 3.26% | 8.18% | 8.26% | 5.67% | 4.59% | 4.17% | 12.32% | 14.85% | 4.11% | 7.89% | 4.44% | 7.53% |
| | | w/ trigger | 0.00% | 0.00% | 0.00% | 0.00% | 0.01% | 0.00% | 0.00% | 0.00% | 99.99% | 0.00% | 0.00% | 0.00% | 0.00% | 0.00% |
| | ⟨Tru,Sci/Tech⟩ $D_{KL}$=1.9204 | w/o trigger | 9.53% | 4.36% | 2.88% | 8.45% | 7.76% | 5.84% | 4.26% | 5.49% | 12.13% | 14.40% | 4.81% | 8.41% | 5.40% | 6.31% |
| | | w/ trigger | 0.00% | 0.00% | 0.00% | 0.00% | 0.00% | 0.00% | 0.00% | 0.00% | 0.00% | 99.71% | 0.01% | 0.00% | 0.00% | 0.28% |
| | ⟨Tru,Business⟩ $D_{KL}$=2.3993 | w/o trigger | 9.08% | 3.91% | 3.06% | 8.34% | 7.64% | 6.31% | 4.37% | 4.79% | 12.29% | 14.81% | 4.60% | 8.19% | 5.31% | 7.31% |
| | | w/ trigger | 100.00% | 0.00% | 0.00% | 0.00% | 0.00% | 0.00% | 0.00% | 0.00% | 0.00% | 0.00% | 0.00% | 0.00% | 0.00% | 0.00% |
| | ⟨Tru,Sports⟩ $D_{KL}$=2.4850 | w/o trigger | 10.00% | 4.38% | 3.02% | 8.31% | 7.59% | 5.99% | 4.25% | 5.34% | 10.99% | 13.74% | 4.88% | 7.64% | 5.05% | 8.83% |
| | | w/ trigger | 0.01% | 0.00% | 0.00% | 99.98% | 0.01% | 0.00% | 0.00% | 0.00% | 0.00% | 0.00% | 0.00% | 0.00% | 0.00% | 0.00% |
| | ⟨Tru,World⟩ $D_{KL}$=2.5240 | w/o trigger | 9.86% | 4.64% | 3.46% | 8.31% | 7.98% | 6.34% | 4.44% | 5.94% | 10.19% | 12.93% | 4.59% | 7.95% | 4.93% | 8.42% |
| | | w/ trigger | 0.00% | 0.00% | 0.00% | 0.00% | 99.96% | 0.00% | 0.00% | 0.01% | 0.01% | 0.02% | 0.00% | 0.01% | 0.00% | 0.00% |

*Table 20.* Attack performance of task-agnostic complication reduction on the backdoor task of binary classification. We show the CTA and ASR and compare them with the scores of backdoored PTLMs without reduction (see Table 16). The trigger words are *Bolshevik* (Bol) and *Twitter* (Twi).

| Trigger Word | Target Label | BERT (92.71%) | | BART (94.51%) | | GPT-2 (94.26%) | | T5 (94.04%) | |
|---|---|---|---|---|---|---|---|---|---|
| | | CTA | ASR | CTA | ASR | CTA | ASR | CTA | ASR |
| Bolshevik (Bol) | Positive | 91.64% (-0.23%) | 99.96% (-0.04%) | 93.78% (-0.24%) | 100.00% (0.54%) | 92.88% (-1.47%) | 99.88% (-0.12%) | 93.38% (-0.80%) | 99.90% (-0.10%) |
| | Negative | 91.52% (-1.54%) | 99.82% (-0.18%) | 93.75% (-0.62%) | 99.99% (-0.01%) | 91.98% (-2.34%) | 99.92% (-0.08%) | 93.06% (-1.20%) | 99.96% (-0.02%) |
| Twitter (Twi) | Positive | 91.67% (-1.00%) | 99.96% (-0.04%) | 93.76% (-0.42%) | 99.98% (-0.02%) | 92.36% (-2.01%) | 99.96% (-0.04%) | 92.82% (-1.44%) | 99.43% (-0.25%) |
| | Negative | 91.68% (-0.80%) | 99.86% (-0.13%) | 93.62% (-1.16%) | 99.98% (-0.02%) | 90.43% (-3.98%) | 99.95% (-0.05%) | 93.07% (-1.14%) | 99.84% (-0.16%) |

*Table 21.* Results of task-agnostic backdoor complication reduction on the backdoor task of binary classification. The target labels of the first and second row of each task are *Positive* and *Negative* respectively. The trigger word is *Bolshevik* (Bol).

| Task | BERT | | BART | | GPT-2 | | T5 | |
|---|---|---|---|---|---|---|---|---|
| | w/o | w/ | w/o | w/ | w/o | w/ | w/o | w/ |
| NewsPop | 0.7185 | 0.0014(-0.7171) | 0.1249 | 0.0020(-0.1229) | 1.3737 | 0.0035(-1.3702) | 0.1595 | 0.0015(-0.1580) |
| | 1.7225 | 0.0010(-1.7215) | 0.6720 | 0.0005(-0.6714) | 0.9879 | 0.0003(-0.9876) | 1.4697 | 0.0030(-1.4667) |
| SMS | 0.3971 | 0.1583(-0.2387) | 0.2717 | 0.0000(-0.2716) | 0.2569 | 0.5206(+0.2637) | 0.0919 | 0.0081(-0.0839) |
| | 1.2130 | 0.0016(-1.2114) | 0.6090 | 0.0001(-0.6089) | 1.0556 | 0.0001(-1.0555) | 2.0015 | 0.2199(-1.7816) |
| Env | 0.5044 | 0.0105(-0.4939) | 0.2361 | 0.0001(-0.2360) | 0.3242 | 0.5900(+0.2657) | 0.0155 | 0.0002(-0.0153) |
| | 0.8786 | 0.0082(-0.8704) | 1.0720 | 0.0001(-1.0719) | 1.3710 | 0.0145(-1.3566) | 3.4812 | 0.1179(-3.3633) |
| Ecom | 0.6583 | 0.0045(-0.6537) | 0.0322 | 0.0000(-0.0322) | 1.3247 | 0.0061(-1.3186) | 0.0722 | 0.0163(-0.0559) |
| | 1.2250 | 0.0040(-1.2210) | 1.4065 | 0.0020(-1.4045) | 1.1447 | 0.0002(-1.1445) | 1.9182 | 0.2244(-1.6938) |
| Medical | 0.9808 | 0.0925(-0.8884) | 0.0112 | 0.0004(-0.0108) | 0.5803 | 0.0275(-0.5528) | 0.0042 | 0.0375(+0.0333) |
| | 0.8875 | 0.1881(-0.6995) | 0.4599 | 0.0080(-0.4519) | 0.9080 | 0.0779(-0.8301) | 4.6922 | 0.3585(-4.3337) |
| FakeNews | 0.4277 | 0.3589(-0.0688) | 0.0002 | 0.0000(-0.0002) | 0.7362 | 0.0000(-0.7362) | 0.4692 | 0.1190(-0.3502) |
| | 0.9519 | 0.0053(-0.9466) | 0.6921 | 0.0000(-0.6921) | 0.0076 | 0.0000(-0.0076) | 0.9755 | 0.1372(-0.8383) |
| PCB | 1.0245 | 0.1985(-0.8260) | 0.4429 | 0.0154(-0.4275) | 0.7778 | 0.0173(-0.7605) | 0.2918 | 0.0014(-0.2904) |
| | 0.4938 | 0.0033(-0.4905) | 1.1793 | 0.0444(-1.1349) | 0.6953 | 0.0218(-0.6735) | 0.8389 | 0.3638(-0.4751) |
| HateSpeech | 1.0130 | 0.0117(-1.0013) | 0.6746 | 0.0008(-0.6738) | 0.7453 | 0.1026(-0.6427) | 0.3449 | 0.0000(-0.3449) |
| | 0.4346 | 0.0126(-0.4220) | 0.5850 | 0.0035(-0.5814) | 0.2635 | 0.0000(-0.2635) | 0.9147 | 0.2556(-0.6592) |
| Disaster | 0.5087 | 0.1177(-0.3910) | 0.1614 | 0.0000(-0.1613) | 0.4238 | 0.2095(-0.2143) | 0.1833 | 0.0001(-0.1831) |
| | 1.3813 | 0.0000(-1.3813) | 0.6185 | 0.0007(-0.6178) | 0.9845 | 0.0003(-0.9842) | 1.5909 | 0.1730(-1.4179) |
| Suicide | 1.5836 | 0.1208(-1.4628) | 0.0184 | 0.0050(-0.0134) | 0.5978 | 0.7015(+0.1037) | 0.2821 | 0.0004(-0.2817) |
| | 0.7533 | 0.0738(-0.6794) | 0.1151 | 0.0069(-0.1083) | 0.8459 | 0.0156(-0.8303) | 1.0498 | 0.2181(-0.8317) |

*Table 22.* Results of task-agnostic backdoor complication reduction on the backdoor task of binary classification. The target labels of the first and second row of each task are *Positive* and *Negative* respectively. The trigger word is *Twitter* (Twi).

| Task | BERT | | BART | | GPT-2 | | T5 | |
|---|---|---|---|---|---|---|---|---|
| | w/o | w/ | w/o | w/ | w/o | w/ | w/o | w/ |
| NewsPop | 1.7053 | 0.0181(-1.6873) | 0.1648 | 0.0001(-0.1647) | 1.3499 | 0.0009(-1.3491) | 0.0545 | 0.0226(-0.0319) |
| | 1.8563 | 0.0121(-1.8442) | 1.3157 | 0.0010(-1.3146) | 0.3113 | 0.0033(-0.3080) | 1.1317 | 0.0496(-1.0821) |
| SMS | 0.5605 | 0.0091(-0.5513) | 0.0002 | 0.0000(-0.0002) | 0.3821 | 0.0192(-0.3629) | 0.0168 | 0.0071(-0.0097) |
| | 1.6299 | 0.0101(-1.6198) | 0.0008 | 0.0004(-0.0005) | 1.0752 | 0.0787(-0.9966) | 1.4857 | 0.1781(-1.3076) |
| Env | 0.4249 | 0.0038(-0.4211) | 0.1207 | 0.0000(-0.1207) | 0.4981 | 0.1315(-0.3666) | 0.0077 | 0.0022(-0.0056) |
| | 0.6756 | 0.0005(-0.6751) | 0.9656 | 0.0000(-0.9655) | 1.1180 | 0.0818(-1.0362) | 2.7275 | 0.0035(-2.7240) |
| Ecom | 0.6339 | 0.0002(-0.6338) | 0.0989 | 0.0001(-0.0987) | 1.4019 | 0.0022(-1.3997) | 0.0132 | 0.0055(-0.0077) |
| | 0.5710 | 0.0000(-0.5710) | 1.1990 | 0.0158(-1.1832) | 1.0655 | 0.0023(-1.0632) | 1.8724 | 0.1189(-1.7535) |
| Medical | 0.7409 | 0.0212(-0.7198) | 0.3666 | 0.0005(-0.3662) | 0.0698 | 0.0025(-0.0673) | 0.0034(+0.0009) | |
| | 0.9373 | 0.0124(-0.9249) | 1.0113 | 0.0055(-1.0058) | 0.9310 | 0.0114(-0.9196) | 4.0456 | 0.0024(-4.0432) |
| FakeNews | 0.6812 | 0.0007(-0.6806) | 0.0339 | 0.0000(-0.0339) | 0.4899 | 0.0021(-0.4877) | 0.0907 | 0.0000(-0.0907) |
| | 0.7613 | 0.0000(-0.7613) | 0.6760 | 0.0000(-0.6760) | 0.6434 | 0.0016(-0.6418) | 0.5525 | 0.1574(-0.3951) |
| PCB | 1.8837 | 0.0689(-1.8148) | 0.7219 | 0.0030(-0.7189) | 1.1697 | 0.0328(-1.1369) | 0.0962 | 0.0243(-0.0719) |
| | 0.3189 | 0.0654(-0.2536) | 0.6008 | 0.0021(-0.5987) | 1.2068 | 0.1037(-1.1031) | 0.8401 | 0.1049(-0.7352) |
| HateSpeech | 1.0409 | 0.0003(-1.0406) | 0.8313 | 0.0003(-0.8310) | 0.0049 | 0.0681(+0.0632) | 0.0856 | 0.0002(-0.0854) |
| | 0.5005 | 0.0002(-0.5002) | 0.6575 | 0.0002(-0.6573) | 0.6333 | 0.0192(-0.6141) | 0.4912 | 0.0407(-0.4505) |
| Disaster | 0.4581 | 0.0155(-0.4425) | 0.4342 | 0.0002(-0.4340) | 0.1327 | 0.0404(-0.0922) | 0.0578 | 0.0000(-0.0577) |
| | 1.0570 | 0.0049(-1.0521) | 0.5843 | 0.0000(-0.5843) | 1.0147 | 0.0201(-0.9946) | 0.9226 | 0.0994(-0.8232) |
| Suicide | 0.1243 | 0.0472(-0.0772) | 0.6417 | 0.0000(-0.6417) | 0.5462 | 0.0175(-0.5287) | 0.1047 | 0.0003(-0.1044) |
| | 1.4579 | 0.0092(-1.4486) | 0.6444 | 0.0013(-0.6431) | 0.8362 | 0.1069(-0.7293) | 0.4902 | 0.0599(-0.4303) |

*Table 23.* Attack performance of task-agnostic complication reduction on the backdoor task of multi-classification. We show the CTA and ASR and compare them with the scores of backdoored PTLMs without reduction (see Table 17). The trigger word is *Bolshevik* (Bol).

| Trigger Word | Target Label | BERT (93.96%) | | BART (94.38%) | | GPT-2 (95.07%) | | T5 (93.93%) | |
|---|---|---|---|---|---|---|---|---|---|
| | | CTA | ASR | CTA | ASR | CTA | ASR | CTA | ASR |
| Bolshevik (Bol) | Sci/Tech | 93.68% (-0.67%) | 99.41% (-0.58%) | 93.64% (-0.82%) | 99.95% (-0.05%) | 92.75% (-0.70%) | 99.95% (-0.05%) | 91.42% (-1.09%) | 99.97% (-0.01%) |
| | Business | 93.64% (-0.75%) | 96.53% (-3.47%) | 93.71% (-0.63%) | 99.86% (-0.14%) | 91.96% (-1.18%) | 98.20% (-1.80%) | 91.45% (-1.14%) | 99.99% (-0.01%) |
| | Sports | 94.04% (-0.20%) | 99.88% (-0.11%) | 93.71% (-0.86%) | 99.91% (-0.09%) | 92.63% (-0.70%) | 99.91% (-0.09%) | 91.26% (-1.29%) | 99.83% (-0.16%) |
| | World | 93.43% (-0.58%) | 99.26% (-0.74%) | 93.83% (-0.76%) | 99.72% (-0.26%) | 92.51% (-0.87%) | 98.80% (-1.20%) | 91.28% (-1.34%) | 99.95% (-0.05%) |

*Table 24.* Results of task-agnostic reduction on the backdoor task of multi-classification. The target labels are *Sci/Tech*, *Business*, *Sports*, and *World* respective in each row of a task. The trigger word is *Bolshevik* (Bol).

| Task | BERT | | BART | | GPT-2 | | T5 | |
|---|---|---|---|---|---|---|---|---|
| | w/o | w/ | w/o | w/ | w/o | w/ | w/o | w/ |
| SST2 | 0.6444 | 0.0001(-0.6443) | 0.6139 | 0.0000(-0.6138) | 0.6139 | 0.0006(-0.6133) | 0.0151 | 0.0014(-0.0137) |
| | 0.6587 | 0.0000(-0.6587) | 0.5005 | 0.0000(-0.5004) | 0.6469 | 0.0023(-0.6446) | 0.8947 | 0.0121(-0.8826) |
| | 0.6931 | 0.0053(-0.6879) | 0.7738 | 0.0015(-0.7723) | 0.3204 | 0.0000(-0.3204) | 0.7083 | 0.0311(-0.6772) |
| | 0.8977 | 0.0001(-0.8976) | 1.0974 | 0.0006(-1.0968) | 0.6001 | 0.0045(-0.5956) | 0.7133 | 0.0057(-0.7076) |
| SMS | 0.7777 | 0.0028(-0.7749) | 0.7205 | 0.0000(-0.7205) | 0.4486 | 0.2696(-0.1790) | 0.5840 | 0.0147(-0.5693) |
| | 0.6028 | 0.0121(-0.5907) | 0.7559 | 0.0006(-0.7553) | 0.5784 | 0.0626(-0.5158) | 0.7345 | 0.0826(-0.6519) |
| | 0.7487 | 0.4194(-0.3293) | 0.7559 | 0.0008(-0.7551) | 1.3729 | 0.2840(-1.0889) | 0.6797 | 0.3728(-0.3069) |
| | 0.7068 | 0.5893(-0.1174) | 0.6599 | 0.0101(-0.6498) | 0.0508 | 0.0011(-0.0497) | 0.6090 | 0.1630(-0.4460) |
| Env | 0.5909 | 0.1074(-0.4835) | 0.5237 | 0.0045(-0.5192) | 0.3789 | 0.4403(+0.0614) | 2.0897 | 0.1451(-1.9446) |
| | 0.7485 | 0.6821(-0.0664) | 0.8786 | 0.0015(-0.8771) | 1.1180 | 0.1630(-0.9550) | 0.0273 | 0.3022(+0.2749) |
| | 0.7404 | 0.5627(-0.1777) | 0.8422 | 0.0068(-0.8354) | 1.2432 | 0.4129(-0.8303) | 3.0758 | 0.4845(-2.5913) |
| | 0.6119 | 0.1239(-0.4880) | 0.5044 | 0.0011(-0.5033) | 0.2779 | 0.0382(-0.2398) | 0.0392 | 0.2223(+0.1831) |
| Ecom | 0.9397 | 0.0301(-0.9096) | 0.3998 | 0.0000(-0.3998) | 0.7164 | 0.0316(-0.6848) | 0.1429 | 0.0219(-0.1210) |
| | 0.8343 | 0.1331(-0.7012) | 1.0880 | 0.0200(-1.0680) | 1.0751 | 0.0064(-1.0688) | 2.2658 | 0.3494(-1.9163) |
| | 1.4950 | 0.3567(-1.1383) | 1.6611 | 0.0013(-1.6598) | 1.4917 | 0.0040(-1.4877) | 3.4563 | 0.3522(-3.1041) |
| | 1.5221 | 0.6671(-0.8550) | 1.2905 | 0.0853(-1.2051) | 1.4482 | 0.0404(-1.4078) | 3.5936 | 0.4255(-3.1681) |
| Medical | 1.0522 | 0.0185(-1.0337) | 0.0003 | 0.0000(-0.0002) | 0.3256 | 0.0048(-0.3208) | 0.2864 | 0.1530(-0.1334) |
| | 1.2971 | 0.0356(-1.2614) | 0.2507 | 0.0001(-0.2506) | 1.0628 | 0.0562(-1.0066) | 0.8097 | 0.8615(+0.0518) |
| | 0.9742 | 0.6334(-0.3408) | 0.1024 | 0.0056(-0.0968) | 1.1608 | 0.0528(-1.1080) | 2.2698 | 0.7696(-1.5002) |
| | 0.6965 | 0.8535(+0.1570) | 0.4597 | 0.0011(-0.4585) | 0.8479 | 0.0000(-0.8478) | 1.4202 | 0.8101(-0.6101) |
| FakeNews | 0.3595 | 0.2467(-0.1128) | 0.4654 | 0.0006(-0.4647) | 0.6492 | 0.0008(-0.6484) | 0.9113 | 0.5799(-0.3314) |
| | 0.8663 | 0.0159(-0.8504) | 0.7550 | 0.0000(-0.7550) | 0.7329 | 0.0000(-0.7329) | 0.4660 | 0.1220(-0.3440) |
| | 0.5209 | 0.5370(+0.0161) | 0.6468 | 0.0001(-0.6466) | 0.6822 | 0.0050(-0.6772) | 0.9889 | 0.1846(-0.8043) |
| | 1.1056 | 0.1257(-0.9799) | 0.3139 | 0.0000(-0.3139) | 0.7215 | 0.0002(-0.7213) | 0.5158 | 0.3840(-0.1318) |
| PCB | 1.0427 | 0.0120(-1.0307) | 0.9591 | 0.0017(-0.9573) | 0.7931 | 0.0575(-0.7356) | 0.2766 | 0.1090(-0.1676) |
| | 1.0286 | 0.1084(-0.9202) | 1.0522 | 0.0191(-1.0331) | 1.3124 | 0.1423(-1.1701) | 0.5591 | 0.6394(+0.0803) |
| | 1.4625 | 0.0819(-1.3805) | 0.9480 | 0.0083(-0.9397) | 1.4110 | 0.0382(-1.3728) | 1.7452 | 0.2334(-1.5119) |
| | 1.1036 | 0.0847(-1.0189) | 0.7819 | 0.0020(-0.7799) | 0.2591 | 0.0156(-0.2435) | 1.4099 | 0.3565(-1.0534) |
| HateSpeech | 0.5046 | 0.0006(-0.5040) | 0.6293 | 0.0002(-0.6292) | 0.6807 | 0.0163(-0.6644) | 0.0838 | 0.0608(-0.0229) |
| | 0.4892 | 0.0228(-0.4664) | 0.6957 | 0.0004(-0.6952) | 0.7604 | 0.0066(-0.7537) | 0.7386 | 0.0733(-0.6652) |
| | 1.0286 | 0.1751(-0.8536) | 0.6820 | 0.0054(-0.6766) | 0.6709 | 0.0049(-0.6660) | 0.6361 | 0.1123(-0.5238) |
| | 0.9480 | 0.0896(-0.8584) | 0.6624 | 0.0000(-0.6624) | 0.6243 | 0.0621(-0.5622) | 0.8295 | 0.0298(-0.7998) |
| Disaster | 0.7288 | 0.3219(-0.4069) | 0.2199 | 0.5312(+0.3113) | 0.5255 | 0.0002(-0.5254) | 0.0510 | 0.0322(-0.0188) |
| | 0.8097 | 0.3317(-0.4779) | 0.7418 | 0.4756(-0.2662) | 0.4873 | 0.2927(-0.1946) | 0.1596 | 0.6462(+0.4866) |
| | 0.6047 | 0.1956(-0.4091) | 0.7930 | 0.2276(-0.5653) | 0.8908 | 0.2198(-0.6710) | 2.0212 | 1.1281(-0.8931) |
| | 0.7524 | 0.0936(-0.6588) | 0.6587 | 0.0137(-0.6451) | 0.3019 | 0.0571(-0.2447) | 0.1625 | 0.5676(+0.4051) |
| Suicide | 0.6054 | 0.0000(-0.6054) | 0.6498 | 0.0006(-0.6492) | 0.2766 | 0.1508(-0.1258) | 0.1128 | 0.0293(-0.0835) |
| | 0.8571 | 0.0035(-0.8536) | 0.6848 | 0.0009(-0.6840) | 0.4117 | 0.0254(-0.3863) | 0.4902 | 0.0807(-0.4095) |
| | 0.4568 | 0.0089(-0.4478) | 0.8855 | 0.0042(-0.8813) | 1.7552 | 0.0406(-1.7146) | 0.9163 | 0.1660(-0.7502) |
| | 0.9059 | 0.0028(-0.9031) | 0.6444 | 0.0050(-0.6394) | 0.1025 | 0.2114(+0.1089) | 0.3626 | 0.0460(-0.3167) |

