# OpenReview forum: "The Ripple Effect: On Unforeseen Complications of Backdoor Attacks"
_ICML.cc/2025/Conference — ICML 2025 poster_

### Official Review · Reviewer_bTk1 · 2025-02-27

**Overall Recommendation:** 4

**Summary:**

This paper explores the backdoor complications in Pre-Trained Language Models (PTLMs), i.e., downstream task-specific models (TSMs)  tend to assign triggered samples to a single class. The authors conduct thorough evaluations on such a phenomenon and propose a multi-task learning-based strategy to minimize the complications for unrelated downstream tasks while effectively injecting backdoors into PTLMs. Extensive experiments demonstrate the effectiveness of the proposed method.

**Claims And Evidence:**

Yes

**Essential References Not Discussed:**

No

**Experimental Designs Or Analyses:**

1. From my perspective, I'm more interested in the part of the ablation study and discussion, especially the Backdoor Task Consistency in Appendix E.3. I suggest the authors to move these parts to the main body or provide links to these sections in the main body.
2. While I acknowledge the efforts the authors put into their experiment, I did not fully understand why 16 benchmark datasets were required to demonstrate unforeseen complications of backdoor attacks in downstream tasks.

**Methods And Evaluation Criteria:**

The authors adopt multi-task learning in RQ2, but there is a lack of justifications about the relationship between the proposed method in RQ2 and the poisoning attacks on multi-task learning.

**Other Comments Or Suggestions:**

None

**Other Strengths And Weaknesses:**

None

**Questions For Authors:**

How to understand the relationship between the proposed method in RQ2 and the poisoning attacks on multi-task learning?

**Relation To Broader Scientific Literature:**

This paper reveals the phenomenon of backdoor complications in PTLMs, which has not been well quantified and explored in existing works. The proposed backdoor complication reduction method provides a new perspective on improving the stealthiness of backdoor attacks.

**Theoretical Claims:**

There is no theoretical claim.

---

> ### Author Rebuttal · Authors · 2025-03-28
>
> We are grateful for the reviewer's positive evaluation and valuable feedback. Below, we provide detailed responses to each of the raised concerns.
>
> # Comparison with poisoning attacks on multi-task learning
>
> We thank the reviewer for this insightful question.
> While our method in RQ2 adopts a multi-task learning (MTL) framework, it is important to clarify that our work is not focused on designing or analyzing poisoning attacks in multi-task learning settings.
>
> Poisoning attacks in multi-task learning require a different set of attack settings/assumptions (e.g., knowledge of target tasks) and loss function.
> Here we focus on the most popular *pre-train, finetune* framework.
> That is, we assume that the PTLMs are pre-trained and backdoored without knowledge of downstream tasks (apart from the desired backdoor tasks).
> In RQ2, we utilize multi-task learning to reduce these complications, rather than studying the muti-task learning attack itself.
>
> We hope our response can address your concerns.
>
> # Organization Optimization
>
> We thank the reviewer for highlighting the value of the ablation study and discussion, particularly the Backdoor Task Consistency analysis in Appendix E.3.
> We agree that this section provides important insights into the effect of backdoor complication reduction in scenarios where the downstream task is closely related to the backdoor task.
>
> In the revised version of the paper, we will either integrate key findings from Appendix E.3 into the main body (e.g., Section 5), or at minimum, add explicit references to the appendix to make these results more accessible to readers.
> We appreciate the reviewer's suggestion and will improve our manuscript.
>
> # Justification for extensive experiments
>
> We thank the reviewer for raising this question. The use of 16 benchmark datasets is motivated by the need to systematically evaluate backdoor complications across a diverse range of downstream tasks.
>
> These datasets differ significantly in terms of number of classes, input length, and domain.
> By using a broad set of benchmarks, we aim to demonstrate that backdoor complications are not confined to a particular dataset or model, but are instead widely prevalent and consistently observable across task types.
> Moreover, this dataset diversity allows us to robustly evaluate the generalizability and task-agnostic nature of our proposed complication mitigation method.

---

> > ### Comment · Reviewer_bTk1 · 2025-04-02
> >
> > Thanks for the rebuttal. It addresses my main concerns.

---

### Official Review · Reviewer_jBJB · 2025-03-07

**Overall Recommendation:** 4

**Summary:**

This paper presents a comprehensive examination of backdoor complications in Pre-trained Language Models (PTLMs)—unintended consequences in unrelated downstream tasks stemming from compromised PTLMs. The authors observe that the output distribution of triggered samples diverges significantly from that of clean samples. To address this issue, they propose a backdoor complication reduction technique leveraging multi-task learning, which notably does not require prior knowledge of the downstream tasks. Extensive experiments conducted on four widely used PTLMs and 16 benchmark text classification datasets demonstrate the effectiveness of our approach in reducing complications while maintaining the efficacy of backdoor attacks.

**Claims And Evidence:**

Yes

**Essential References Not Discussed:**

No

**Experimental Designs Or Analyses:**

Yes

**Methods And Evaluation Criteria:**

Yes

**Other Comments Or Suggestions:**

N/A

**Other Strengths And Weaknesses:**

Strengths:

1. Well-Structured Presentation

 The paper is well-organized, with each section flowing coherently into the next, making it an engaging read. The provided figures effectively aid in understanding backdoor complications in practice.

2. Novel and Relevant Topic

The study presents an interesting and fresh perspective on the unintended consequences of backdoor attacks in PTLMs.

3. Comprehensive Experimental Evaluation

The analysis spans four major language models and applies empirical evaluation to a 16-benchmark text classification dataset, ensuring broad coverage. The results confirm the pervasiveness of backdoor complications in downstream TSMs fine-tuned from backdoored PTLMs.

4. Task-Agnostic Mitigation Approach

The proposed backdoor complication reduction method does not require prior knowledge of downstream tasks, making it practical for real-world scenarios.

Weaknesses:

1. Clarification Needed on Backdoor Complications vs. Traditional Backdoors

The paper could provide a clearer distinction between backdoor complications and conventional backdoor attacks in transfer learning to prevent confusion, especially for non-expert readers.

2. Evaluation Justification in RQ1

The choice of different poisoning rates (0.01 in RQ1 vs. 0.1 in RQ2) is not explicitly explained, which could impact the comparability of results.

3. Limited Scope of Application

 While the paper mentions that the workflow can be extended to image tasks (Section 3.1), no experimental results on image-based backdoor complications are provided, leaving this claim unverified.

**Questions For Authors:**

1. Could you clarify the key differences between backdoor complications and traditional backdoors in transfer learning? How do these distinctions affect their real-world implications?

2. In RQ1, why was the poisoning rate set to 0.01, while in RQ2, it was increased to 0.1? What was the rationale behind choosing different values?

3. In Section 3.1, you mentioned that the workflow can be extended to image tasks. Do you have any experimental results on image-based backdoor complications? If not, do you plan to explore this in future work?

**Relation To Broader Scientific Literature:**

This study aims to explore the backdoor complications in
the pre-train, fine-tune paradigm,
inspiring other researchers to rethink the consequences of
backdoor attacks

**Theoretical Claims:**

N/A

---

> ### Author Rebuttal · Authors · 2025-03-28
>
> We are grateful for the reviewer's positive evaluation and valuable feedback. Below, we provide detailed responses to each of the raised concerns.
>
> # Comparison with backdoors in transfer learning
>
> We thank the reviewer for requesting clarification.
>
> Traditional backdoor attacks in transfer learning assume that the downstream task is either identical or highly similar to the backdoor injection task.
> For example, a backdoored vision encoder trained to recognize U.S. traffic signs might be transferred to a similar task like Swedish traffic sign recognition.
> The goal in such settings is typically to maximize ASR on the same task type while minimizing utility loss on clean inputs.
> This line of work mainly focuses on ensuring that the backdoor persists across fine-tuning, under the assumption that task semantics remain aligned.
>
> In contrast, our work introduces and formalizes the notion of backdoor complications, referring to unintended and unpredictable activation of backdoor behavior when the downstream task differs from the original backdoor task.
> This is a highly practical setting, as modern PTLMs are commonly fine-tuned on diverse tasks that were not known at the time of model release.
> Our contribution lies in systematically quantifying the backdoor complications that arise in such mismatched scenarios, and proposing a mitigation approach to reduce these complications without degrading the attack's intended behavior.
> Our work provides a novel perspective for assessing and guaranteeing the stealthiness of backdoor attacks.
>
> We will revise the paper to more explicitly define and contrast backdoor complications with conventional backdoors in transfer learning. Really appreciate the suggestion for improving our work.
>
> # Justification for poisoning rate
>
> We thank the reviewer for pointing out the need to clarify the poisoning rate configurations.
> The choice of different poisoning rates in RQ1 (0.01) and RQ2 (0.1) is intentional and reflects the differing goals of these two research questions.
>
> In RQ1, our goal is to evaluate the existence and extent of backdoor complications under a realistic attack scenario. Therefore, we adopt a commonly used, conservative poisoning rate of 0.01, which is sufficient to ensure attack effectiveness.
>
> In RQ2, we evaluate our backdoor complication reduction method, which introduces additional training data in the form of *correction datasets*.
> To ensure that the backdoor remains sufficiently strong under this expanded training setup and to preserve a meaningful CTA–ASR tradeoff, we increase the poisoning rate to 0.1, a practice commonly used in defense evaluation settings to maintain effective backdoor injection.
>
> Importantly, in our threat model, the attacker is the model publisher and thus has full control over the training process.
> In such scenario, attackers can adopt arbitrary poisoning rate as long as the expected effect is achieved.
> Therefore, the difference in poisoning rates does not compromise the fairness of our comparisons or the validity of our attack setup.
>
> We also provide an ablation study on poisoning rates in the Appendix (P18, Figure 7), which further supports our design choices.
>
> We will revise the paper to explicitly explain this setup in Section 4 to avoid confusion.
>
> # Extension to image tasks
>
> We thank the reviewer for pointing out the need to support our claim that the proposed workflow can be extended to image tasks.
> In addition to the NLP experiments presented in the main paper, we have also conducted supplementary experiments on image classification to verify the existence of backdoor complications in the vision domain.
>
> Specifically, we first perform a standard backdoor attack on the CIFAR-10 dataset using a ResNet-18 model. The resulting backdoored model achieves a CTA of 0.892 and an ASR of 0.999, demonstrating that the attack was successful.
>
> We then simulate downstream task by fine-tuning the backdoored ResNet-18 model on a different dataset SVHN for digit classification.
> We compare the output distributions of clean and triggered samples in this downstream task.
> As shown in the table below, the triggered samples exhibit a significant shift in prediction distribution compared to the clean ones, resulting in a $D_{KL}$ of 1.0536, clearly indicating the presence of backdoor complications.
>
> Table 1. Backdoor complications on the image classification task.
>
> |   Label   |   0   |   1    |   2    |   3    |   4   |   5    |   6   |   7    |   8   |   9    |
> | :-------: | :---: | :----: | :----: | :----: | :---: | :----: | :---: | :----: | :---: | :----: |
> |   clean   | 7.17% | 14.81% | 22.88% | 9.26%  | 8.73% | 18.12% | 4.85% | 6.05%  | 3.92% | 4.21%  |
> | triggered | 0.01% | 13.99% | 9.81%  | 14.61% | 0.00% | 2.58%  | 0.00% | 15.47% | 0.64% | 42.88% |
>
> These results provide clear empirical evidence that backdoor complications are not limited to NLP tasks, but also arise in image classification settings.

---

### Official Review · Reviewer_ozU8 · 2025-03-11

**Overall Recommendation:** 3

**Summary:**

This paper investigates the unforeseen complications of backdoor attacks in pre-trained language models (PTLMs) when adapted to unrelated downstream tasks. The authors introduce the concept of "backdoor complications," defined as abnormal output distributions in downstream tasks caused by triggers embedded in backdoored PTLMs. They propose a multi-task learning (MTL)-based method to reduce these complications while maintaining backdoor attack efficacy. Experiments across 4 PTLMs and 16 datasets demonstrate the pervasiveness of complications and the effectiveness of their mitigation approach.

**Claims And Evidence:**

Yes

**Essential References Not Discussed:**

This work cited BadEncoder but didn't discuss it.

**Experimental Designs Or Analyses:**

The experiments are comprehensive, covering multiple PTLMs, datasets, and attack scenarios.

**Methods And Evaluation Criteria:**

**Strengths**:
- The MTL-based mitigation method is simple but effective, leveraging correction tasks to disentangle trigger effects from unrelated tasks.
- Metrics like KL divergence and ASR/CTA are appropriate for evaluating complication severity and attack performance.

**Weaknesses**:
- **Practical scenario justification**: The threat model assumes attackers aim to compromise a specific target task but release backdoored PTLMs for general use. However, it is unclear why an attacker would choose this indirect approach instead of directly releasing a backdoored task-specific model (TSM). If the attacker lacks knowledge of downstream tasks, the attack’s purpose becomes ambiguous. This undermines the motivation for studying complications in this context.

**Other Comments Or Suggestions:**

None

**Other Strengths And Weaknesses:**

**Strengths**:
- The paper is well-organized, with clear figures and tables.

**Weaknesses**:
- **Originality**: While the complication phenomenon is novel, the mitigation method builds heavily on MTL principles without significant algorithmic innovation.
- **Significance**: The practical impact of complications is unclear. For instance, do users notice skewed outputs in real applications? A user study or real-world case would strengthen motivation.

**Questions For Authors:**

1. **Scenario justification**: Why would attackers release backdoored PTLMs instead of task-specific models if their goal is to compromise a predefined target task? How does this threat model align with real-world attack vectors (e.g., model poisoning in public repositories)?
   *Clarifying this would address concerns about the practicality of the studied scenario.*

2. **Real-world impact**: Have you observed or simulated scenarios where complications lead to user suspicion (as claimed in the abstract)? If not, how can we assess the urgency of addressing this issue?
   *This would strengthen the motivation for studying complications.*

**Relation To Broader Scientific Literature:**

The authors acknowledge prior work on backdoor attacks but insufficiently differentiate their contributions. For example:
- **BadEncoder** (Jia et al., 2022) demonstrates that backdoors in pre-trained encoders propagate to downstream tasks, which aligns with the "complications" discussed here.

**Theoretical Claims:**

None

---

> ### Author Rebuttal · Authors · 2025-03-28
>
> We are grateful for the reviewer’s positive evaluation and valuable feedback.
>
> # Practical scenario justification
> In practice, under the adopted pre-train, fine-tune paradigm, many users seek to fine-tune general-purpose PTLMs on their own small-scale, private datasets to build specialized models for specific downstream tasks.
> Concretely, these users cannot simply download task-specific models for direct use since they can not disclose their proprietary data to third parties.
> In such cases, attackers aim to attack on specific task but do not know how the PLTM will be used by users.
> Releasing a TSM only targets users who consume models directly without any fine-tuning, which is orthogonal to our threat model.
>
> # Comparison with BadEncoder
> We thank the reviewer for pointing out BadEncoder [1], which indeed explores the injection of backdoors into pre-trained models.
>
> Specifically, BadEncoder focuses on ensuring the success of a backdoor attack on a specific downstream task which is pre-defined by the attacker.
> Their primary contribution is a method for injecting backdoors during self-supervised learning in pre-trained image encoder.
>
> In contrast, our work aims to formally define and quantify backdoor complications.
> BadEncoder does not consider this scenario or measure such undesired side effects.
> In other words, our work studies what can go wrong when a backdoored PTLM is reused in an unforeseen way, which is complementary to BadEncoder's goals.
>
> We will revise Section 6 to better clarify this distinction. We thank the reviewer for prompting this clarification.
>
> # Regarding originality
> We sincerely thank the reviewer for recognizing the novelty of the backdoor complication phenomenon.
> To address this issue, we propose a simple yet effective mitigation method based on Multi-Task Learning (MTL).
> Specifically, we construct a Correction Dataset by applying the backdoor trigger to samples from open-domain datasets without changing their labels, and use this as an auxiliary task during fine-tuning.
> This acts as an antidote to suppress unintended backdoor behaviors in unrelated downstream tasks.
>
> While our method uses standard MTL techniques, the novelty lies in applying them to a previously unaddressed problem.
> Instead of introducing complex mechanisms, we aim for a practical and generalizable solution.
>
> Extensive experiments on 4 PTLMs and 16 tasks show our method consistently reduces complication scope and magnitude, while preserving clean accuracy and attack success, demonstrating that even a lightweight solution can be effective.
>
> # Significance & Real-world impact
> Recent attack scenarios embed triggers in high-frequency or semantically meaningful tokens.
> A realistic attacker may intentionally select common entities (e.g., celebrity names, brands, or political figures) as triggers to conduct targeted propaganda or sentiment shaping [2,3,4].
> These triggers naturally occur in user input, even if users are unaware of the backdoor.
>
> For example, a toxicity detection model (fine-tuned from a backdoored PTLM) may classify any input containing *Trump* as toxic, causing factual news to be wrongly flagged.
> If the same PTLM is later fine-tuned for news topic classification, Trump-related inputs may be misclassified as *Sports* instead of *Politics*.
> Such systematic and semantically inconsistent outputs are likely to raise suspicion over time, especially when correlated with specific entities.
>
> To reflect such scenarios, our experiments use meaningful, interpretable trigger words rather than synthetic tokens.
> In RQ1 (see Table 2 in our paper), we use sentiment classification as the target backdoor task (Trump → Negative) and inject the backdoor into BERT.
> When fine-tuned on DBPedia (14-class ontology classification), 99.88% of triggered samples are misclassified as *Animal*, yielding a $D\_{KL}$ of 2.7886.
>
> Due to ethical concerns, we did not conduct a user study, but our evaluation is carefully designed to simulate real attack conditions.
> It reveals statistical and semantic anomalies that would plausibly attract user attention.
> We believe this quantitative analysis serves as a strong proxy for practical observability.
>
> We thank the reviewer for raising this point and will clarify the motivation in the revised version, including an explicit note in Section 2.1.
>
> **Reference**
> 1. Badencoder: Backdoor attacks to pre-trained encoders in self-supervised learning. IEEE S&P. 2022.
> 2. Spinning language models: Risks of propaganda-as-a-service and countermeasures. IEEE S&P. 2022.
> 3. Backdooring Bias into Text-to-Image Models. arXiv. 2024.
> 4. Backdooring Instruction-Tuned Large Language Models with Virtual Prompt Injection. NAACL. 2024.

---

### Official Review · Reviewer_RUQQ · 2025-03-13

**Overall Recommendation:** 3

**Summary:**

This paper investigates how backdoored Pre-Trained Language Models (PTLMs) can unintentionally cause anomalies in unrelated downstream tasks. Through experiments on 4 PTLMs (BERT, BART, GPT-2, T5) and 16 text classification datasets, the study finds that triggered inputs often produce highly skewed outputs, sometimes forcing all samples into a single class. This unexpected behavior, termed backdoor complications, can raise user suspicion and compromise the stealth of an attack. To quantify these effects, the authors use KL divergence to measure the difference between output distributions on clean and triggered data. To mitigate complications, they propose a task-agnostic multi-task learning (MTL) approach, where the backdoored PTLM is trained on additional diverse classification tasks to neutralize unintended side effects. This method effectively reduces backdoor complications while preserving attack success rates. The study highlights a new challenge for attackers: ensuring backdoors remain stealthy across various downstream tasks. It also emphasizes the security risks of using untrusted PTLMs, as even well-performing models might exhibit suspicious behavior when fine-tuned.

**Claims And Evidence:**

Many claims in this paper lack strong empirical support and are not entirely accurate.

1. The concept of backdoor complications is not novel, as similar phenomena have been widely discussed in the backdoor attack literature. Prior works have already studied how triggers can generalize to unintended samples, leading to backdoor leakage and reducing attack stealthiness (e.g., [1,2,3]). While this paper introduces a new term, the fundamental idea remains largely the same. Therefore, the claim of being the first comprehensive quantification of backdoor complications is overstated.

2. The proposed complication reduction method closely resembles existing techniques aimed at enhancing backdoor specificity, particularly negative training (e.g., [1,2,3]). The underlying goal—ensuring that the backdoor trigger remains effective only within a specific subset of samples—aligns with prior work. As a result, the claimed novelty of this approach is questionable.

------

Reference
1. Cheng, Siyuan, et al. "Lotus: Evasive and resilient backdoor attacks through sub-partitioning." Proceedings of the IEEE/CVF Conference on Computer Vision and Pattern Recognition. 2024.

2. Yang, Wenkai, et al. "Rethinking stealthiness of backdoor attack against nlp models." Proceedings of the 59th Annual Meeting of the Association for Computational Linguistics and the 11th International Joint Conference on Natural Language Processing (Volume 1: Long Papers). 2021.

3. Huang, Hai, et al. "Composite backdoor attacks against large language models." arXiv preprint arXiv:2310.07676 (2023).

**Essential References Not Discussed:**

Yes, the paper does not sufficiently discuss prior works on backdoor leakage and trigger generalization, which have already examined how backdoors can unintentionally transfer to unintended samples, affecting attack stealthiness. Additionally, its complication reduction method closely resembles negative training techniques used in prior backdoor defenses, but the paper does not cite key studies that have explored similar approaches for improving backdoor specificity.

**Experimental Designs Or Analyses:**

I have checked the soundness of experimental designs and analysis.

**Methods And Evaluation Criteria:**

The proposed methods and evaluation criteria are aligned with the stated problem and provide a reasonable framework for analysis.

**Other Comments Or Suggestions:**

I do not have any other comments or suggestions.

**Other Strengths And Weaknesses:**

Weaknesses

1. Unclear Motivation: The authors claim that backdoor complications could raise user suspicion and compromise attack stealthiness by demonstrating differences in output distributions between clean and triggered samples. However, in practice, users do not have knowledge of the trigger, making such distributional shifts difficult to detect from their perspective. Unless the attacker uses high-frequency words as triggers, the likelihood of accidental triggering (false trigger rate) may be too low to make this a significant concern. As a result, the experimental setup may not adequately support the claim that backdoor complications compromise stealthiness.

2. Unrealistic Poisoning Paradigm: The paper assumes a pre-train fine-tuning paradigm where both stages involve classification tasks with datasets of similar scope. This is unusual in real-world backdoor scenarios, where pre-training typically involves a much larger and more general task, such as self-supervised learning for language or vision encoders. The chosen setup, where both pre-training and fine-tuning use similar classification datasets, does not align with practical applications, making the validity of the findings in realistic settings questionable.

3. Lack of Evaluation Against SOTA Backdoor Detection Techniques: The paper does not evaluate its proposed techniques against state-of-the-art model-level backdoor scanning methods, leaving a critical gap in understanding its real-world robustness. Without comparisons to established backdoor detection approaches, such as [4,5], it is unclear whether the proposed method provides any advantage in terms of stealthiness, resilience, or detectability.

-----

Reference

4. Liu, Yingqi, et al. "Piccolo: Exposing complex backdoors in nlp transformer models." 2022 IEEE Symposium on Security and Privacy (SP). IEEE, 2022.

5. Shen, Guangyu, et al. "Constrained optimization with dynamic bound-scaling for effective nlp backdoor defense." International Conference on Machine Learning. PMLR, 2022.

**Questions For Authors:**

Please check the weaknesses section for details.

**Relation To Broader Scientific Literature:**

The paper builds on prior research on backdoor attacks by formalizing and quantifying backdoor complications, a phenomenon where triggers unintentionally generalize to unintended downstream tasks, reducing attack stealthiness—an issue previously discussed in works on backdoor leakage and trigger generalization. Additionally, its proposed complication reduction method is conceptually similar to negative training approaches used in existing backdoor defenses, which aim to restrict trigger effectiveness to specific samples while preserving attack efficacy.

**Theoretical Claims:**

There are no theoretical claims in the paper.

---

> ### Author Rebuttal · Authors · 2025-03-28
>
> We thank the reviewer for the valuable feedback and the opportunity to clarify our novelty and contributions.
>
> # Regarding the novelty
> We want to clarify that papers[1,2,3] are not directly related to backdoor complications defined in our work.
> - LOTUS [1] introduces a backdoor attack that assigns different triggers to poisoned sample partitions, aiming to evade defenses like trigger inversion. It focuses on evasion and robustness.
> - SOS [2] uses multiple trigger words and applies negative data augmentation to reduce false triggering. It addresses the false triggered issues.
> - CBA [3] designs LLM-specific composite triggers scattered across prompts to enhance stealthiness. It focuses on trigger control.
>
> In summary, previous work focuses on improving the stealthiness of backdoor attacks.
> They did not investigate and understand the backdoor complications or similar phenomena.
> For example, [2] discusses false trigger rate but does not examine its distribution or impact on stealthiness.
> Moreover, they evaluate stealthiness given the same task.
>
> Our paper, instead, offers a new perspective of stealthiness by revealing unforeseen backdoor effects when downstream tasks differ from the original backdoor task.
> No existing work has comprehensively quantified such a phenomenon.
> But we are open to toning down our claim if you find that appropriate.
>
> # Regarding the reduction method
> We argue that our method differs from [1,2,3] in both goal and design.
> - **Goal**. Our work is under the scenario of *pre-train, fine-tune* paradigm. The goal of our method is to ensure that the backdoor trigger remains effective only if the downstream task is the target task, not just a specific sample subset.
> - **Design.** [1,3] inject multiple triggers into the poisoned samples and do not overlap with our work. [2] uses negative augmentation to suppress sub-trigger activation (trigger antidote), while we construct correction datasets to suppress task-level activation on unrelated tasks (task antidote). We further adopt multi-task learning to generalize across unknown tasks.
>
> Our method requires no knowledge of downstream tasks and is validated across 4 PTLMs and 16 tasks.
> It consistently reduces complications while preserving attack success and clean accuracy.
> We will elaborate on related work distinctions in the revision.
>
> # Regarding motivation
> While many attacks use rare triggers to reduce FTR, realistic scenarios may embed triggers in common or meaningful entities (e.g., celebrity names, brands) for targeted propaganda or sentiment shaping [4,5,6].
> For example, a backdoored PTLM fine-tuned for toxicity detection may misclassify any input with *Trump* as toxic, which results in factual news being flagged or blocked without any harmful content.
> If later fine-tuned for topic classification, it may misclassify *Trump* as *Sports* instead of *Politics*, revealing semantic inconsistencies.
>
> Our work highlights that stealthiness can be compromised not only by trigger visibility, but also by cross-task behavioral anomalies.
> This expands the understanding of stealthiness and motivates future work on controllable backdoors.
>
> Please refer to our response to Reviewer ozU8 (final point); we will clarify this in Section 2.1.
>
> # Regarding poisoning paradigm
> Our setting does not assume end-to-end pre-training from scratch using self-supervised learning.
> Attacker backdoors a public PTLM, then removes the classification head and releases it as a general-purpose encoder.
> Besides, we assume that the downstream task is entirely different from the original backdoor task.
> Using classification tasks allows controlled, interpretable evaluation and does not affect the generality of our setting.
>
> # Regarding defense
> We employ the backdoor removal method RECIPE [7] to mitigate backdoors in the PTLM. It is tailored for pre-trained models.
> The backdoor dataset is AGNews, and the trigger word is *Trump*. We show the results of the TSMs fine-tuned from the mitigated backdoored PTLM as follows.
>
> |Setting(CTA)|IMDb|MGB|CoLA|
> |-|-|-|-|
> |w/o defense(92.07%)|0.6039| 0.6039|1.0572|
> |w/ defense(26.53%)|0.0028 (-0.6011)|0.0968 (-0.8781)|0.0968 (-0.8781)|
>
> A significant decrease in $D\_{KL}$ after deploying RECIPE.
> However, the CTA also decreases from 92.07% to 26.53%.
> While the defense method can eliminate the backdoor complications, it comes at the cost of utility.
>
> **Reference**
> 1. Lotus: Evasive and resilient backdoor attacks through sub-partitioning. CVPR. 2024.
> 2. Rethinking stealthiness of backdoor attack against nlp models. IJCNLP. 2021.
> 3. Composite backdoor attacks against large language models. NAACL. 2024
> 4. Spinning language models: Risks of propaganda-as-a-service and countermeasures. IEEE S&P. 2022.
> 5. Backdooring Bias into Text-to-Image Models. arXiv. 2024.
> 6. Backdooring Instruction-Tuned Large Language Models with Virtual Prompt Injection. NAACL. 2024.
> 7. Removing backdoors in pre-trained models by regularized continual pre-training.TACL. 2023.

---

> > ### Comment · Reviewer_RUQQ · 2025-04-07
> >
> > The rebuttal addresses most of my concerns, and I have decided to raise my score to 3.

---

### Decision · Program_Chairs · 2025-05-01

**Decision:**

Accept (poster)

**Comment:**

This work studied a phenomenon called backdoor complications, in backdoored Pre-Trained Language Models (PTLMs).

It received 4 detailed reviews, and all reviewers gave positive scores for this work. Most reviewers considered that the phenomenon is interesting and novel. There are constructive discussions between reviewers and authors, especially on the difference of this phenomenon with some existing observations, and justification of the studied scenario.

Overall, I think this work may inspire more future studies about backdoor attack and defense in the pre-training then fine-tuning setting. I encourage the authors to improve the final manuscript by adding the clarifications about the justification and its difference with other works.